# Automated detection and segmentation of non-small cell lung cancer computed tomography images

Sergey P. Primakov [1], Abdalla Ibrahim [1,2,3,4,5], Janita E. van Timmeren [1,6], Guangyao Wu[1,7], Simon A. Keek[1], Manon Beuque [1], Renée W. Y. Granzier [8], Elizaveta Lavrova [1,9], Madeleine Scrivener[10], Sebastian Sanduleanu[1], Esma Kayan[1], Iva Halilaj [1], Anouk Lenaers [1,8], Jianlin Wu[11], René Monshouwer [12], Xavier Geets[10], Hester A. Gietema [2], Lizza E. L. Hendriks [13], Olivier Morin [14], Arthur Jochems[1], Henry C. Woodruff [1,2] & Philippe Lambin [1,2✉]

Detection and segmentation of abnormalities on medical images is highly important for patient management including diagnosis, radiotherapy, response evaluation, as well as for quantitative image research. We present a fully automated pipeline for the detection and volumetric segmentation of non-small cell lung cancer (NSCLC) developed and validated on 1328 thoracic CT scans from 8 institutions. Along with quantitative performance detailed by image slice thickness, tumor size, image interpretation difficulty, and tumor location, we report an in-silico prospective clinical trial, where we show that the proposed method is faster and more reproducible compared to the experts. Moreover, we demonstrate that on average, radiologists & radiation oncologists preferred automatic segmentations in 56% of the cases. Additionally, we evaluate the prognostic power of the automatic contours by applying RECIST criteria and measuring the tumor volumes. Segmentations by our method stratified patients into low and high survival groups with higher significance compared to those methods based on manual contours.

[1] The D-Lab, Department of Precision Medicine, GROW- School for Oncology and Reproduction, Maastricht University, Maastricht, The Netherlands. [2] Department of Radiology and Nuclear Medicine, GROW - School for Oncology and Reproduction, Maastricht University Medical Centre+, Maastricht, The Netherlands. [3] Division of Nuclear Medicine and Oncological Imaging, Department of Medical Physics, Hospital Center Universitaire De Liege, Liege, Belgium. [4] Department of Nuclear Medicine and Comprehensive diagnostic center Aachen (CDCA), University Hospital RWTH Aachen University, Aachen, Germany. [5] Department of Radiology, Columbia University Irving Medical Center, New York, USA. [6] Department of Radiation Oncology, University Hospital Zürich and University of Zürich, Zürich, Switzerland. [7] Department of Radiology, Union Hospital, Tongji Medical College, Huazhong University of Science and Technology, Wuhan, China. [8] Department of Surgery, GROW - School for Oncology and Reproduction, Maastricht University Medical Centre+, Maastricht, The Netherlands. [9] GIGA Cyclotron Research Centre In Vivo Imaging, University of Liège, Liège, Belgium. [10] Department of Radiation Oncology, Cliniques universitaires St-Luc, Brussels, Belgium. [11] Department of Radiology, Affiliated Zhongshan Hospital of Dalian University, Dalian, China. [12] Department of Radiation Oncology, Radboud University Medical Center, Nijmegen, The Netherlands. [13] Department of Pulmonary Diseases, GROW - School for Oncology and Reproduction, Maastricht University Medical Center, Maastricht, the Netherlands. [14] Department of Radiation Oncology, University of California San Francisco, San Francisco, California, CA, USA. ✉email: philippe.lambin@maastrichtuniversity.nl

Lung cancer is the deadliest of all cancers afflicting both sexes, accounting for 18.4% of the total cancer deaths worldwide in 2018, almost equal to breast and colon cancers combined[1]. Recent advances in treatment (immune checkpoint inhibitors, tyrosine kinase inhibitors) has significantly improved survival times for subgroups of patients. However, much work is still to be done in the field of lung cancer, especially in screening and early detection. Automated detection and segmentation would immediately impact the clinical workflow in radiotherapy, one of the most common treatment modalities for lung cancer[2]. Radiotherapy uses medical imaging, especially computed tomography (CT), to obtain accurate tumor localization and electron densities for the purpose of treatment planning dose calculations[3]. Accurate segmentation of the tumor and organs at risk are also essential as errors might lead to over- or under-irradiation of both the tumor and/or healthy tissue. It has been estimated that a 1 mm shift of the tumor segmentation could affect the radiotherapeutic dose calculations by up to 15%[4,5]. Therefore, automated accurate segmentation can significantly reduce the time needed by clinicians to carryout treatment planning, and adaptive re-planning of treatment depending on the changes in the tumor.

Equally important are the lesion and organ at risk segmentation process for radiation oncologists for radiotherapy planning, and the measurement of lesions within the Response Evaluation Criteria in Solid Tumors (RECIST) 1.1 framework for radiologists, both laborious manual routines which impose an avoidable workload[6]. Currently, such segmentations and appropriate RECIST measurements are performed manually or semi-automatically, consuming valuable time and resources, as well as being prone to inter- and intra-observer variability[7].

Another field to profit directly from automated detection and delineation of lesions is radiomics, the high-throughput mining of quantitative features from medical images and their subsequent correlation with clinical and/or biological endpoints[8,9]. Radiomics has the potential to facilitate personalized medicine via diagnostic and predictive models based on phenotypic properties of the region of interest (ROI) being analyzed[10]. ROI segmentation is currently considered to be one of the most time-intensive and laborious steps within the entire radiomics workflow[11].

The recent advancement of machine learning techniques, combined with improvements in the quality and archiving of medical images, have fueled intensive research in the field of artificial intelligence (AI) for medical imaging analysis[12,13]. Deep learning, a branch of AI-based artificial neural networks, has been successfully applied on images to solve problems such as classification or segmentation[14,15]. Several attempts have been made to adapt these methods for medical imaging problems, including tumor detection and segmentation on CT images[16–19]. A major hurdle in developing fully automated software that can be applied to any CT is the heterogeneity of the datasets, especially when acquired from multiple centers[20]. CT scans with different acquisition- or reconstruction parameters present lung structures differently. The methods described in the current literature usually lack a CT preprocessing module in the pipeline, and the problem of data harmonization is left to be solved by a data-driven approach, requiring large datasets representing all aspects of this inhomogeneity.

Taking into consideration these clinical and research needs for lung tumor segmentation, the implementation of automated detection software that is capable of fast and accurate delineation of NSCLC on thoracic CT scans is desirable, bordering on necessity. The applications and benefits include, but are not limited to: (1) CT-based automated screening of lung cancer; (2) Retrospective analysis of entire databases of patients who underwent thoracic CT in daily care for research purposes; (3) Consistent and reproducible segmentations, which are important in planning and monitoring (radio)therapy, and in research; (4) Follow-up of treated primary lung cancer; (5) Automation and acceleration of certain aspects of the clinical radiotherapy workflow, making adaptive re-planning more feasible.

Automated segmentation of NSCLC tumors requires prior identification of the lesion as NSCLC. Invasive tissue biopsy is currently the clinical gold standard in identifying NSCLC. However, an accurate automated segmentation tool requires high detection accuracy. Therefore, software that can automatically segment NSCLC tumors could also be used as a detection method, decreasing the need for invasive biopsies.

In this work, we present a fully automated lung tumor detection and 3D volumetric segmentation pipeline that is capable of handling a large variety of CT acquisition and reconstruction parameters. Furthermore, we externally validate our method on three datasets, compare the volumetric prognostic factor to an existing clinical standard, compare the quantitative performance to a similar published method, and compare the preference score, speed, and reproducibility of our method to those of experts in a prospective clinical trial setting.

## Results

Overall, 1328 thoracic volumetric CT scans with corresponding 3-dimensional tumor segmentations were used in order to train, test, and externally validate a fully automated method for detection and segmentation of NSCLC in standard-of-care images. Datasets 1–7 were combined and randomly divided into training and testing datasets with 999 patients and 93 patients, respectively (see Table 1). Datasets 8–10, comprising 236 patients were used for external validation of the method. A summary of the data is provided in Table 1, description of patient characteristics is provided in Supplementary Table 2.

**Tumor detection and segmentation.** A three-step workflow was developed and successfully implemented (Fig. 1): (i) image pre-processing, a crucial step as datasets collected for this work were obtained from different scanners with various image acquisition and reconstruction protocols (Fig. 1 suppl.). The data inhomogeneity necessitated the harmonization of CT data in order to achieve comparable representations of the tumor region, reduce computational power requirements and image noise, and to optimize contrast; (ii) lung isolation, which allows the model to focus on the ROI and the input of the entire CT scans; (iii) automated tumor detection and segmentation, employing the convolutional neural network.

The ability of the system to detect tumors was assessed lung-wise and yielded a sensitivity of 0.97 and specificity of 0.99 in the external validation dataset and an area under the receiver operating characteristic curve (AUC) of 0.98. Confusion matrices for the detection performance can be found in supplementary materials (Fig. 2 Suppl.). The median contouring performance in the external validation dataset as assessed by the volumetric Dice similarity coefficient (DSC) was 0.82, while the 95th percentile of the Hausdorff distance (H95th) was 9.43 mm. Further metrics, associated uncertainties, as well as test dataset results are reported in Table 2. Using dataset 8 we have established the tolerance level $\tau$ for NSCLC manual segmentation variability ($\tau = 1.18$ mm), allowing the application of the Surface DSC for the NSCLC segmentation task.

Model performance was also separately assessed in regard to groupings of image slice-thickness, tumor size, expert-reported tumor complexity, and tumor location. The sub-cohorts were analyzed for significant differences in model performance, with the results reported in Table 3. As some of the tumors had two or

**Table 1 Description of the datasets used in this study.**

| Ref. | # | Name | Use | Medical Center | #CT scans | #CT scans used | #CT slices with tumor (%) | #CT slices without tumor (%)* | Mean tumor volume (ml) |
|---|---|---|---|---|---|---|---|---|---|
| 40 | 1 | Maastro-CT-Lung-1 | Training/Testing | Open source | 422 | 419 | 4262 (16) | 22490 (84) | 71.0 |
| N/A | 2 | UCL-CT-Lung | Training/Testing | Université catholique de Louvain (TCIA) | 39 | 39 | 400 (16) | 2096 (84) | 53.44 |
| N/A | 3 | UCSF-CT-Lung | Training/Testing/Clinical trial | University of California - San Francisco | 101 | 101 | 689 (11) | 5775 (89) | 19.35 |
| N/A | 4 | MUMC+ Inoperable Lung | Training/Testing | Maastricht University Medical Center+ | 92 | 91 | 1247 (21) | 4577 (79) | 94.99 |
| N/A | 5 | AZHDU Lung | Training/Testing | Affiliated Zhongshan Hospital of Dalian University | 222 | 222 | 464 (4) | 9456 (96) | 2.08 |
| 41 | 6 | Stanford Lung | Training/Testing | Open source (TCIA) | 211 | 137 | 796 (10) | 7396 (90) | 22.37 |
| 42 | 7 | TCIA-CT-Lung-3 | Training/Testing | Open source (TCIA) | 89 | 83 | 630 (12) | 4618 (88) | 51.39 |
| 43 | 8 | The Maastro interobserver reproducibility test | External validation | Open source (BMIA XNAT) | 22 | 20 | 210 (16) | 1070 (84) | 88.03 |
| N/A | 9 | Radboud Lung 2 | External validation | Radboud University Medical Center | 132 | 132 | 3493 (22) | 12460 (78) | 92.04 |
| N/A | 10 | MUMC/Heerlen lung | External validation | MUMC/Heerlen | 89 | 84 | 1120 (13) | 7317 (87) | 77.79 |
| - | - | Overall training/test | - | - | 1176 | 1092 | 8488 (13) | 56408 (87) | 49.07 |
| - | - | Overall validation | - | - | 238 | 236 | 4823 (19) | 20843 (81) | 88.93 |

*CT slices without a segmentation were considered as not containing tumor.

more unconnected components (satellite lesions, or edges of the tumor), the Hausdorff metric can yield unreliable distances when the distance between different volume fragments are calculated. Therefore, the interquartile range (IQR) for H95th was not provided. Histograms showing the distributions of detection and segmentation results are provided in the supplementary materials (Fig. 2 suppl. and Fig. 3 suppl.).

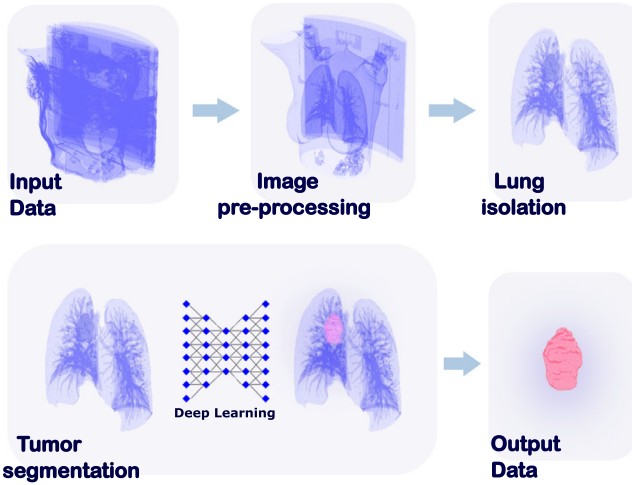

**Fig. 1 Graphic representation of the major steps in the proposed workflow.** Proposed workflow is fully automatic and due to preprocessing step can handle variability in CT scans.

Box plots showing DSC distributions in the sub cohort's tumor size and tumor complexity for both test and validation datasets are shown in Fig. 2. There is a clear trend toward better performance and less variability for larger and less complex tumors. More comparisons for differing slice-thickness groups, complexity classes, tumor location, and tumor sizes performed on the test and external validation dataset are provided in the supplementary materials (Figs. 4–7 suppl.).

Examples of the automatically generated segmentations (from the validation set) in comparison to contours segmented by experts are shown in Fig. 3.

**Comparison to a published method**. A previously published external segmentation model[19] was evaluated on dataset 8 and compared to our model. The performance of the published model was evaluated using two different inputs: (i) as described in the original article (using patches of $256 \times 256$ pixels centered on the tumor); (ii) using the whole slice. For that dataset, our method achieved a DSC of 0.87 (IQR = 0.12), whereas the published method achieved a DSC of 0.83 (IQR = 0.16) when the cropped tumor regions were used and a DSC of 0.09 (IQR = 0.19) in the fully automated configuration (no pre-cropping). Figures for DSC, Ji, and H95th are provided in the supplementary materials (Fig. 8 suppl.).

**Prognostic power of automatic segmentation**. Datasets 1 and 6 were used to compare the prognostic power of measurements extracted from automatically generated and manual contours, as they had available survival data. We calculated the RECIST largest diameter and the tumor volume for both the expert and the

**Table 2 Overview of quantitative model performance.**

| Data, # of patients | Detection performance | | | Segmentation performance | | | | |
|---|---|---|---|---|---|---|---|---|
| | Lung-wise AUC (CI) | Specificity | Sensitivity | DSC (IQR) | Ji (IQR) | H95th, mm | Surf DSC [$\tau = 1.18$] (IQR) | APL, cm |
| Test, 93 | 0.96 (0.94–0.98) | 0.97 | 0.96 | 0.85 (0.15) | 0.74 (0.22) | 5 | 0.75 (0.29) | 106 (274) |
| External validation, 236 | 0.98 (0.97–0.99) | 0.99 | 0.97 | 0.82 (0.17) | 0.70 (0.24) | 9.43 | 0.63 (0.28) | 306 (984) |

*IQR interquartile range, DSC dice similarity coefficient, Ji Jaccard index, H95th 95th percentile, Hausdorff distance.*

**Table 3 Overview of quantitative model performance with regard to various factors.**

| Factors | Test | | | | External Validation | | | |
|---|---|---|---|---|---|---|---|---|
| | DSC (IQR) | Significance | | | DSC (IQR) | Significance | | |
| Slice thickness, 0–2.5 (mm) | 0.86 (0.1) | - | ns | ns | 0.90 (0.08) | - | ns | ns |
| Slice thickness, 2.5–5 (mm) | 0.88 (0.17) | ns | ns | - | 0.81 (0.18) | ** | ns | - |
| Slice thickness, >5 (mm) | 0.83 (0.1) | ns | - | ns | 0.86 (0.13) | ** | - | ns |
| Complexity label, 0 (No PET needed) | 0.88 (0.16) | **** | - | - | 0.87 (0.12) | **** | - | - |
| Complexity label, 1 (PET needed) | 0.84 (0.15) | **** | - | - | 0.79 (0.19) | **** | - | - |
| Tumor size, <20 (ml) | 0.84 (0.11) | - | ns | ns | 0.79 (0.26) | - | ns | ns |
| Tumor size, 20–150 (ml) | 0.86 (0.15) | ns | ns | - | 0.82 (0.16) | * | ns | - |
| Tumor size, >150 (ml) | 0.89 (0.12) | ns | - | ns | 0.86 (0.15) | * | - | ns |
| Tumor location, parenchyma | 0.82 (0.15) | - | ns | ns | 0.83 (0.14) | - | ns ns | ns |
| Tumor location, mediastinum | 0.87 (0.15) | ns | ns | - | 0.80 (0.19) | **** | - | - |
| Tumor location, chest-wall involvement | 0.88 (0.09) | ns | - | ns | 0.89 (0.08) | **** | - | ns |

*Statistical significance were calculated within the factor groups using a two-sided Mann–Whitney–Wilcoxon test with Bonferroni correction wand referred to as follows: "ns" refers to the p value in the range: 5.00e-02 < p ≤ 1.00e+00.*
*\*refers to the p value in the range: 1.00e-02 < p ≤ 5.00e-02; \*\* refers to the p value in the range: 1.00e-03 < p ≤ 1.00e-02; \*\*\* refers to the p value in the range: 1.00e-04 < p ≤ 1.00e-03; \*\*\*\* refers to the p value in the range: p ≤ 1.00e-04.*

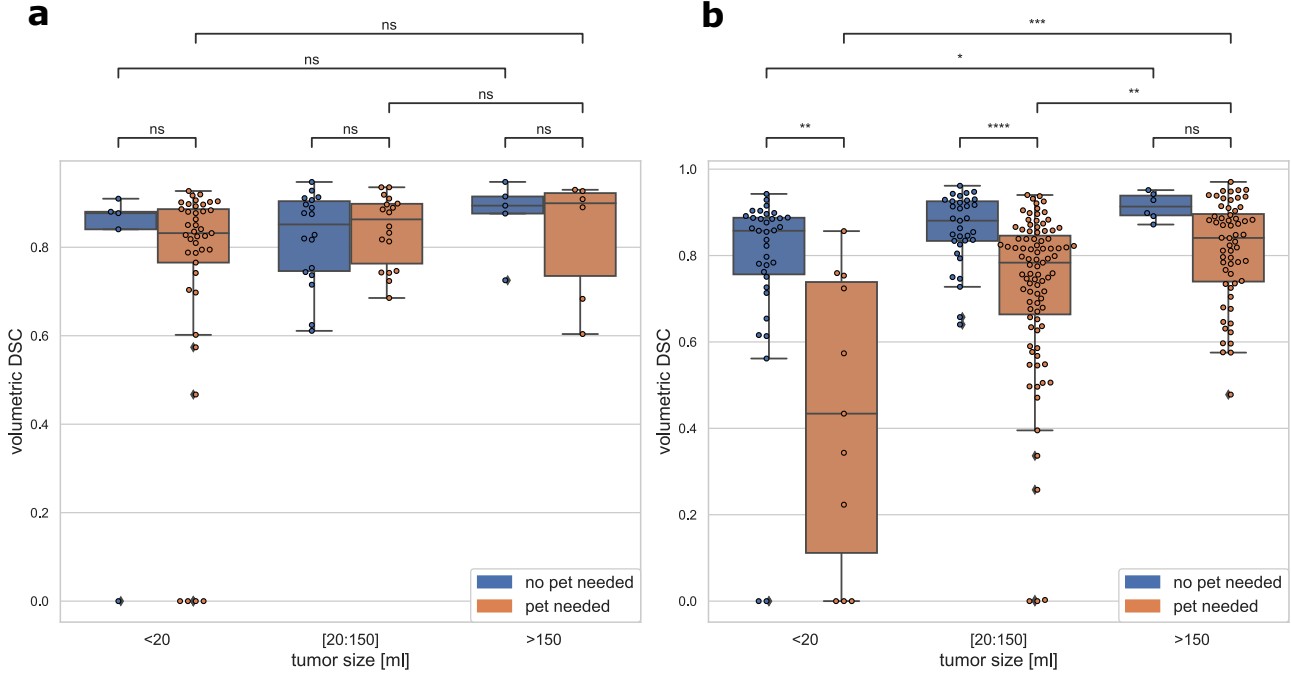

**Fig. 2 Quantitative performance with regards to tumor size and complexity.** Quantitative performance is measured in volumetric dice similarity coefficient (DSC). Tumor complexity is defined through the necessity of using PET to produce segmentation. Data were presented as box plots with overlaid swarm plots, where boxes are representing the interquartile range (IQR), extending from Q1 to Q3 and centered on the median value. Upper whiskers represent the highest data point that is less than $Q3 + 1.5 \times IQR$. Lower whiskers represent the smallest data point that is greater than $Q1 − 1.5 \times IQR$. Data points outside whiskers are considered as outliers. *P* values were calculated using a two-sided Mann–Whitney–Wilcoxon test with Bonferroni correction and referred as follows: "ns" on the plot refers to the *p* value in the range: $5.00e\text{-}02 < p \leq 1.00e\text{+}00$; *refers to the *p* value in the range: $1.00e\text{-}02 < p \leq 5.00e\text{-}02$; **refers to the *p* value in the range: $1.00e\text{-}03 < p \leq 1.00e\text{-}02$; ***refers to the *p* value in the range: $1.00e\text{-}04 < p \leq 1.00e\text{-}03$; ****refers to the *p* value in the range: $p \leq 1.00e\text{-}04$. The exact *p* values are reported in the order from left to right and from the top to the bottom as they are displayed on the figures. Calculations provided for: **a** the test dataset of 93 independent NSCLC CT scans, corresponding *p* values are: 1.000e+00, 1.000e+00, 1.000e+00, 1.000e+00, 1.000e+00, and 1.000e+00; **b** the external validation dataset of 236 independent NSCLC CT scans, corresponding *p* values are: 4.120e-04, 4.022e-02, 8.471e-03, 2.259e-03, 1.662e-05, and 1.117e-01.

automatic segmentation and found that for both metrics the automatically generated segmentations have more prognostic power. Statistical differences in the probability of survival for two groups separated by the median values of these measurements for automated and manual segmentations are reported in Table 4. Kaplan–Meier curves for survival split based on the tumor volume are shown in Fig. 4. KM curves for survival split based on RECIST score can be found in the supplementary materials (Fig. 9 suppl.). Additionally, we have also evaluated the difference using univariate cox analysis to report the cut-off independent results and looked at the scatter plot for tumor volumes. C-index, hazard ratio, and *p* values for a univariate Cox regression are reported in Table 3 in the supplementary materials. Scatter plots for tumor volume based on manual vs automated segmentations can be found in the supplementary materials (Fig. 10 suppl.).

**In silico clinical trial**. A registered in silico clinical trial was performed to assess the following endpoints: (1) the time needed for the processes of manual and automated segmentation; (2) inter and intra-observer variability; (3) the preference of experts for manual or automatically generated segmentations.

For the first and second endpoints, seven medical imaging specialists experienced in NSCLC contouring were asked to contour the tumors of 25 patients from dataset 3 while being timed. Our automated method was significantly faster than the fastest participant ($p < 0.0001$). The mean time for the automated method was 2.78 s/patient (SD = 0.44), whereas the mean time

for manual segmentation was 172.19 s/patient (SD = 158.99) (Fig. 5a).

The median DSC for intra-observer variability among all experts was 0.88 (IQR = 0.12) whereas automated segmentations were 100% reproducible. Individual intra-observer variability scores are reported in Fig. 5b and the JI and H95th are reported in the supplementary materials (Fig. 11a, b suppl.). The median DSC for interobserver variability was 0.81 (IQR = 0.24) (see Fig. 12 suppl.).

The results for assessment of the variability between expert clinicians and the proposed automatic segmentation method achieved on the validation dataset 8 are presented in Fig. 6. Our method achieved an average DSC of 0.82 (IQR = 0.14), whereas the average DSC of experts inter-variability was 0.84 (IQR = 0.12).

For the third endpoint, we had 40 participants from four different backgrounds: four health/medicine master students, 17 computer scientists, 12 medical doctors working in the field of medical imaging, and seven medical specialists (radiologists or radiation oncologists). In order to quantitatively evaluate the qualitative preferences of experts regarding automated vs manual contours, we developed a software tool which allowed experts to visually compare the segmentation and choose their preferences.

On average, the participants preferred the automatic segmentation above the expert's contour in 55% (IQR = 12%) of the cases (Fig. 13a suppl.). Among the groups the qualitative preference scores were as follows: students = 51% (IQR = 4%) computer scientists = 52% (IQR = 14%), medical doctors = 56%

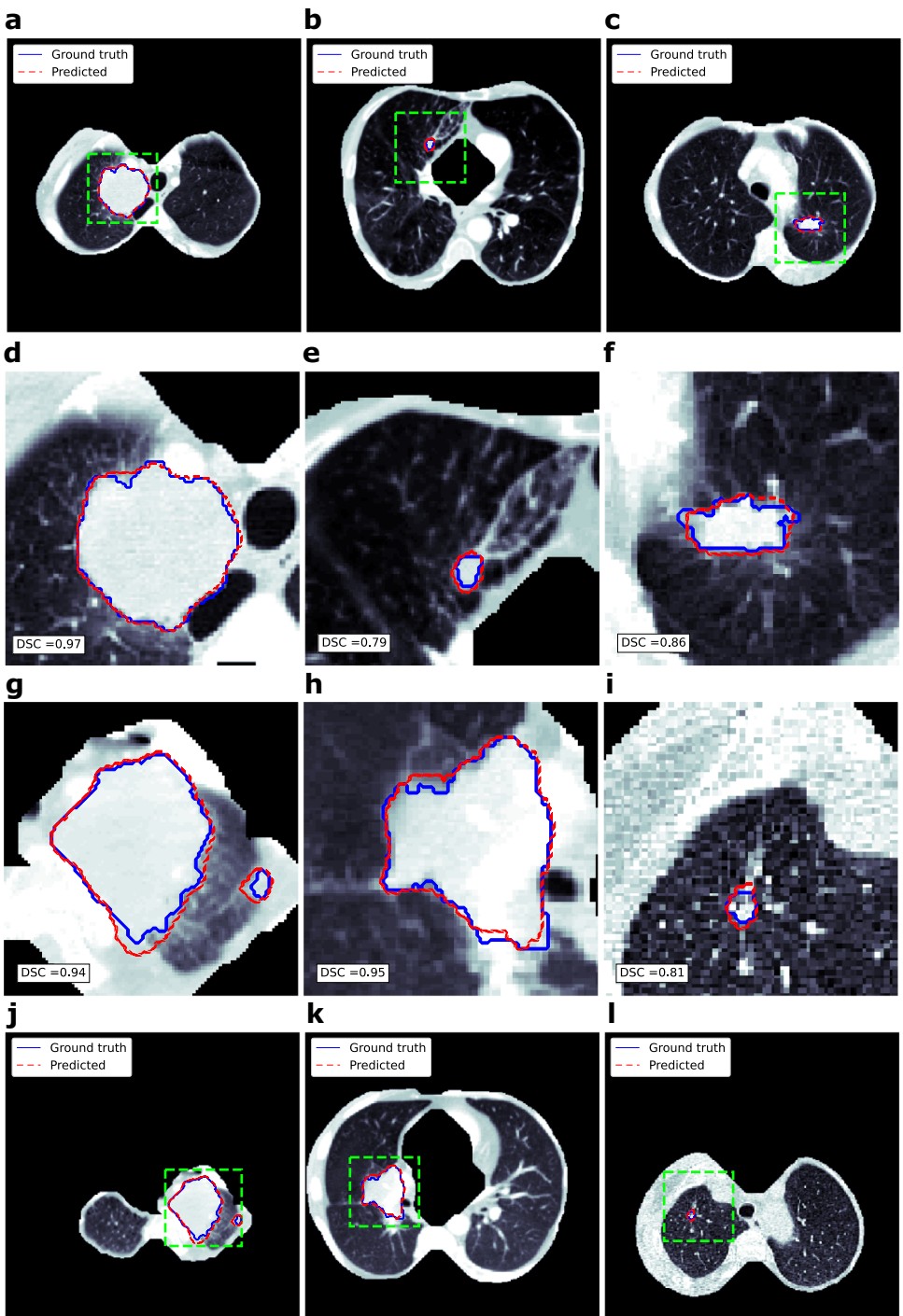

**Fig. 3 Visualization of segmentations.** Automatically generated tumor segmentations are shown as red lines while manual segmentations are shown in blue, the green dashed box shows the area to be magnified for better visuals. **d–i** display magnified area for the (**a–c**, **j–l**) respectively. Corresponding 2D dice similarity coefficient is provided in the bottom left corner on the (**d–i**).

**Table 4 Statistical difference between survival groups separated by the median values of RECIST and tumor volume.**

| Data, (# of patients) | RECIST manual segmentation (p value) | RECIST automatic segmentation (p value) | Tumor volume manual segmentation (p value) | Tumor volume automatic segmentation (p value) |
|---|---|---|---|---|
| 1,419 | 0.0003 | <0.0001 | 0.0017 | <0.0001 |
| 6,137 | 0.0038 | 0.0031 | 0.031 | 0.013 |

Statistical comparisons were performed using log-rank test.

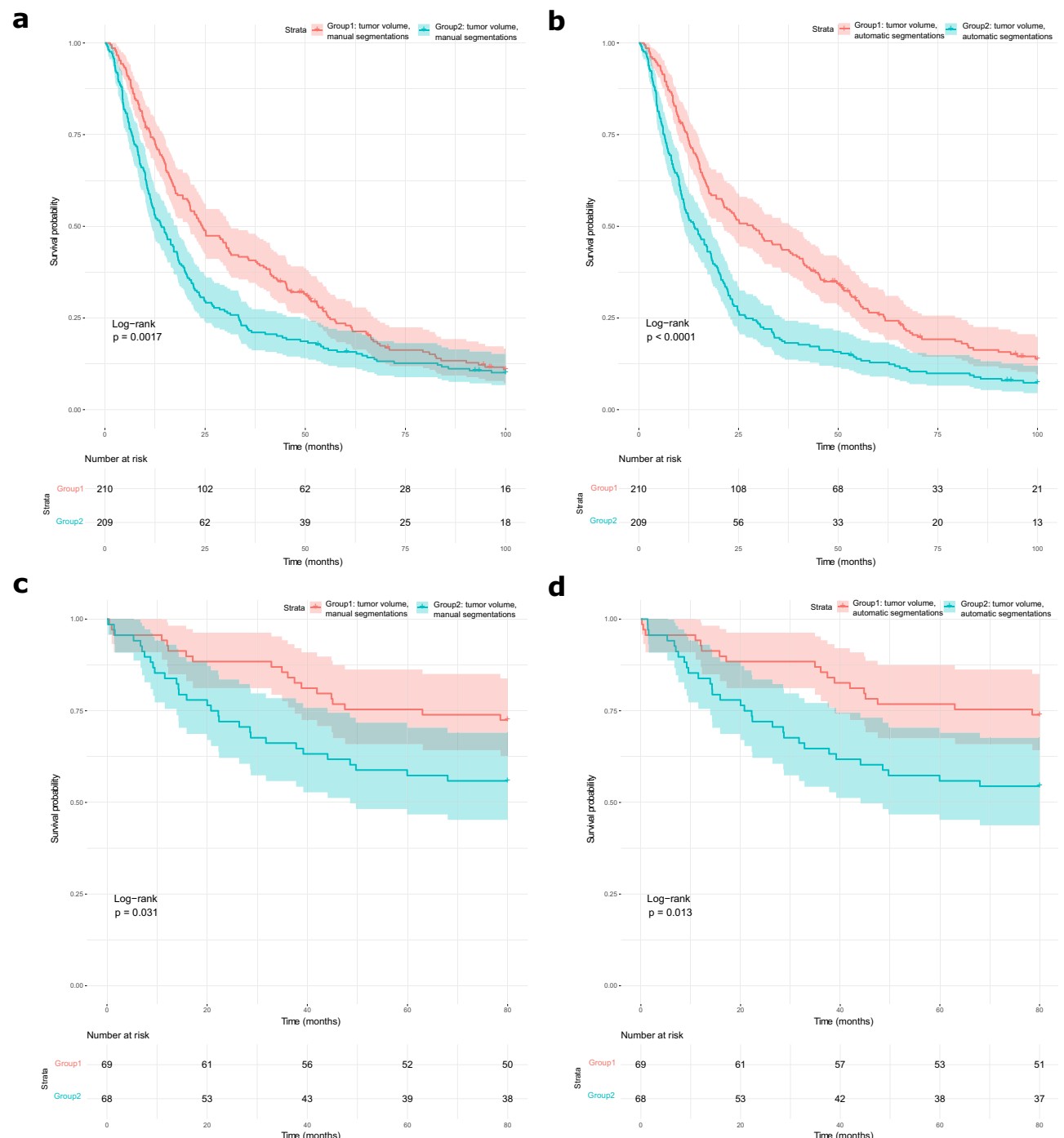

**Fig. 4 Prognostic power of NSCLC segmentations measured with tumor volume.** Comparison of prognostic power of non-small cell lung cancer (NSCLC) segmentation is measured through tumor volume. Tumor volume is calculated based on the manual (**a**, **c**) and automatically generated contours (**b**, **d**). Kaplan–Meyer curves for survival groups based on tumor volume are displayed with 95% pointwise confidence intervals. *P* values are calculated using the log-rank test. Vertical hash marks indicate censored data. **a**, **b** KM curves for Maastro-CT-Lung-1 cohort of 419 NSCLC patients. **c**, **d** KM curves for Stanford Lung cohort of 137 NSCLC patients.

(IQR = 12%) and radiologists and radiation oncologists = 59% (IQR = 13%) (Fig. 13b suppl.).

## Discussion

We presented a deep learning-based approach that is able to achieve state-of-the-art detection and 3D volumetric segmentation of NSCLC on CT scans. Although several attempts to develop lung cancer CT detection and segmentation methods

have been previously made, we believe our work is standing out, especially in its external validation and ability to work on full thoracic CT scans without further input needed by a human operator. To improve detection and segmentation performance, we introduced several complementary steps to the automatic segmentation pipeline: (1) a harmonization routine for the pre-processing of CT scans in order to more comprehensively unify patterns on the images for the models to learn from; (2) a robust computer vision-based method to isolate the lung area, allowing

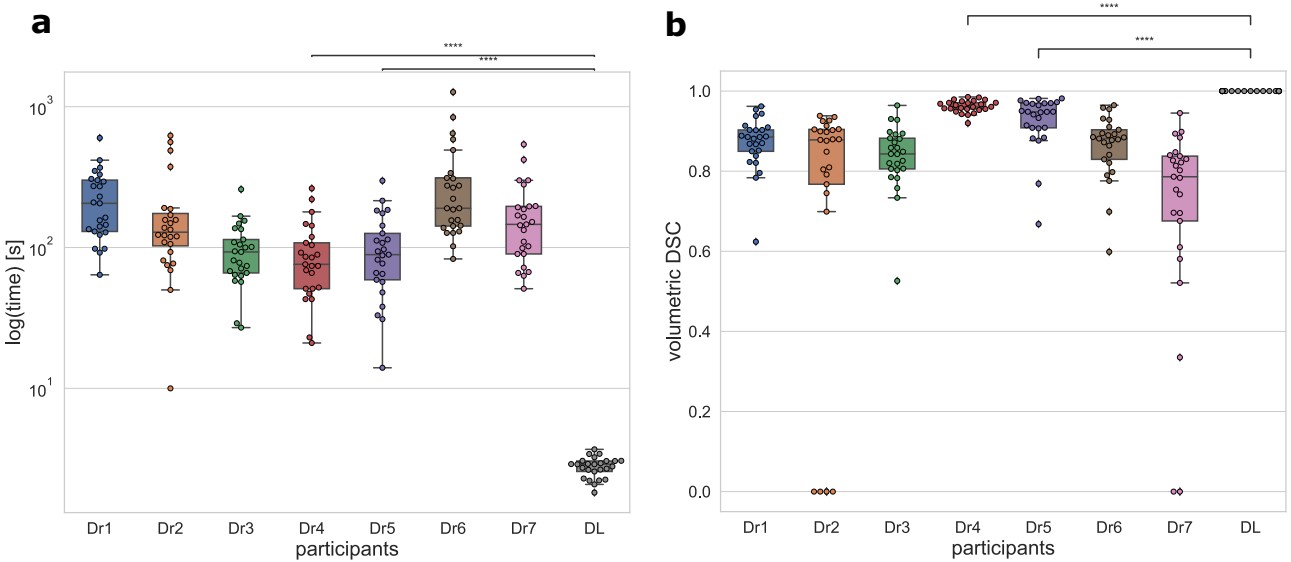

**Fig. 5 Contouring time and intra-observer variability.** Data were presented as box plots with overlaid swarm plots, where boxes are representing the interquartile range (IQR), extending from Q1 to Q3 and centered on the median value. Upper whiskers represent the highest data point that is less than $Q3 + 1.5 \times IQR$. Lower whiskers represent the smallest data point that is greater than $Q1 - 1.5 \times IQR$. Data points outside whiskers are considered outliers. $P$ values were calculated using a two-sided Mann–Whitney–Wilcoxon test with Bonferroni correction and referred as follows: ****refers to the $p$ value in the range: $p \leq 1.00e\text{-}04$. The exact $p$ values are reported in the order from the top to the bottom as they are displayed on the figures. Dr1, Dr2, Dr3, Dr4, Dr5, Dr6, and Dr7—represent contours made by the medical doctors, DL—represents automatically generated contours. **a** Distribution of contouring time was obtained on the 25 NSCLC patients by seven participants and the automated method, corresponding $p$ values are 2.816e-09 and 2.824e-09. **b** Volumetric dice similarity coefficient (DSC) representing intra-observer variability, across participants and the automated method, obtained on the 25 NSCLC patients, corresponding $p$ values are: 1.946e-10 and 1.946e-10.

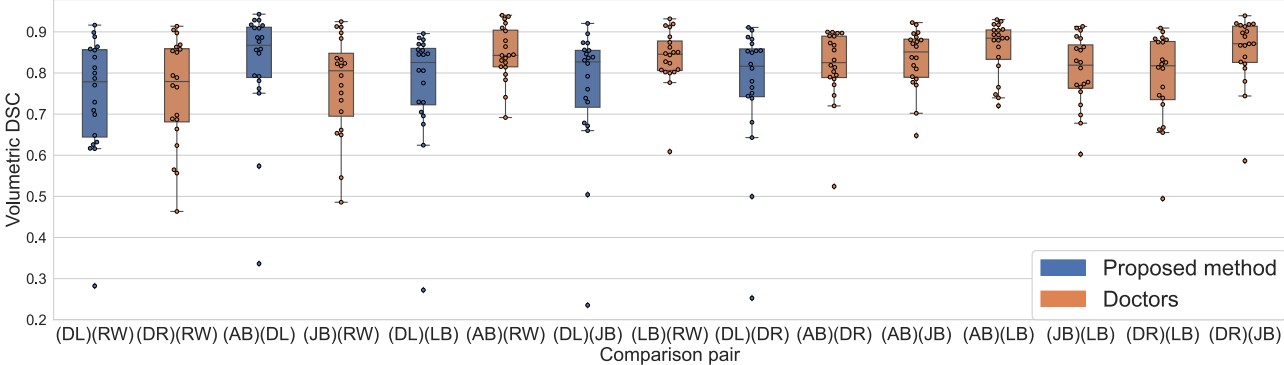

**Fig. 6 Method performance vs interobserver variability.** Quantitative segmentation performance and interobserver variability is measured using volumetric DSC across comparison pairs obtained on 20 NSCLC patients. DR1, DR2, DR3, DR4, and DR5—represent contours made by the doctors (expert clinicians), DL—represents automatically generated contours. Orange box plots correspond to manual segmentation vs manual segmentation comparison and display interobserver variability. Blue box plots correspond to the proposed method vs manual segmentations comparisons and display the proposed method performance. Data were presented as box plots with overlaid swarm plots, where boxes are representing the interquartile range (IQR), extending from Q1 to Q3 and centered on the median value. Upper whiskers represent the highest data point that is less than $Q3 + 1.5 \times IQR$. Lower whiskers represent the smallest data point that is greater than $Q1 - 1.5 \times IQR$. Data points outside whiskers are considered outliers.

the subsequent deep learning step to focus on the region of interest; (3) a dynamically changing loss function for the training procedure, allowing us to control and modify the quality of produced segmentation; (4) CTs of lung abnormalities other than NSCLC were included in the training dataset as negative examples, allowing our method to exclude them from the detection and segmentation process; (5) lung CT slices without contours were also used in the training process as negative samples, thereby increasing the number of unique training samples and decreasing the false-positive rate of the model; (6) although a 2D DL architecture was employed, a 3D post-processing routine produced volumetric segmentation. A prospective, registered in silico clinical trial showed that the performance of the automatic

segmentation model is acceptable by modern clinical standards and that participants preferred automatic segmentations more often than the manual contours. Furthermore, RECIST and tumor volume based on the automatic contours were able to generate a more significant split of survival groups than manual contours.

To set our model in the context of similar published work, Kamal et al. (2018)[17] used a Recurrent 3D-DenseUNet architecture to segment lung cancers which allowed them to obtain a DSC of 0.74 on a validation dataset of 40 patients. Jue et al. (2019)[19] evaluated several 2D convolutional neural network (CNN) architectures such as U-net, Segnet, full-resolution residual neural network (FRRN), and incremental multiple resolution

residual network (MRRN) to segment patches of $160 \times 160$ pixels centered around the tumor, achieving DSC of 0.68 on the external validation dataset. Zhang et al. (2020)[21] used a modified version of ResNet to automatically segment GTV and achieved an averaged dice similarity coefficient (DSC) of 0.73 on the test set, lacking however external validation of the model. Ardila et al. (2019)[16] developed a deep learning-based software, which can detect lung cancer on low dose CTs with an AUC of 94.4%. In our study we were not able to evaluate a patient based AUC for lung cancer detection since all patients had cancer, instead, we have demonstrated that our model was able to detect lungs containing cancer on low dose CTs with a robust AUC of 0.96 in the test and 0.98 in the external validation datasets. Additionally, we evaluated the performance of a published 3D U-net-based approach on our validation dataset, where our model outperformed the published method.

The state-of-the-art detection accuracy and the fact that it accepts any CT containing the lungs as input means the software can be used as a method for screening and detection of lung cancer. This is further corroborated by the fact that CT scans acquired using different parameters can be directly put in, making our method multi-vendor and multi-reconstruction compliant to a certain degree. The inclusion of cases that were hard to segment without a co-registered PET scan allows the deep learning networks to learn how to differentiate tumors from other lung abnormalities such as atelectasis and tumors with mediastinal involvement, which in conjunction with the accurate segmentation of the 3D tumor volume means it can be used clinically in radiotherapy settings or for big data radiomics (and potentially other) research. The robust automatic volumetric and RECIST measurements will subsequently have a positive impact on sample size calculations for clinical trials[22].

Although we attempted to address the flaws and limitations of previous research while developing our software, there were limitations to our work. The ground truth segmentations were originally made on primary NSCLC. Therefore, although the software has a high detection accuracy, it is hypothetically limited to the detection and segmentation of primary NSCLC tumors. Moreover, by considering medical expert contours as the ground truth and taking into account the high interobserver variability of the contouring process[23], the deep learning network was also learning inaccuracies, such as contoured air (that certainly is not cancerous). However, this effect can be alleviated by increasing the training dataset size.

In future work we will utilize the evaluated image factors (slice-thickness, complexity class, predicted tumor size, and tumor location) in order to give a confidence score to each segmentation produced, providing added information to the user about which segmentations might need more attention. Additionally, we think it would be interesting to evaluate our method in a prospective clinical trial setting for tumor response to treatment evaluation utilizing the automatic volumetric RECIST measurement. Since our method was trained only on the planning/pretreatment CT scans, post-treatment changes in the tumor and lung structures may impose extra challenges on our automated segmentation approach.

Further tuning of the model on NSCLC CT scans, and other independent NSCLC datasets can improve the performance of the software, and advance it towards clinical implementation.

The ability of the software developed in this study to handle full thoracic CT scans with different acquisition and reconstruction parameters and without further human intervention represents the pillar for its clinical transition. Clinical application of this software following prospective validation can have a positive impact on the management of lung cancer patients, as it will improve the detection accuracy, and provide a fast, consistent,

and reliable volumetric segmentation for treatment (evaluation) purposes. Furthermore, the use of the software in large radiomics studies will allow automation and will reduce the time needed to complete the studies in a robust manner, as it will significantly decrease the time needed for the rate-limiting part of the workflow—tumor segmentation.

## Methods

**Description of data**. The pretreatment CT scans of 1414 NSCLC patients were retrospectively collected and anonymized by each center and approved by the respective institutional review boards. A description of the data were provided in Table 1, and a description of patient characteristics is provided in Supplementary Table 2.

In this study, which followed the Standards for Reporting of Diagnostic Accuracy Studies statement[24], the requirement for written informed consent was waived. The institutional review board of Maastricht University Medical Center has waived the need for informed consent since the data were anonymized and retrospectively collected with no intervention planned for participants based on the study, and no compensations were provided. The images in dataset 8 were segmented by five radiation oncologists, which allowed us to compare the performance of the deep learning segmentation model to multiple manual delineations. All other segmentations were performed by a radiologist or radiation-oncologist at the center where the diagnosis was made and checked by at least one segmentation expert at our site. The expert segmentations were considered the ground truth for training and further evaluations. Eighty-six patients from various datasets were excluded due to missing tumor contours and the lack of a PET scan to perform the segmentations according to a clinical protocol. Survival data and CT scans for datasets 1 and 6 were collected from the open sources.

**Image preprocessing**. Data inhomogeneity necessitated the harmonization of CT data in order to achieve comparable representations of the tumor region. Furthermore, several steps were introduced to reduce computational power requirements and image noise and to optimize the contrast. The first step is the extraction of a 3D array with voxel intensity values represented as Hounsfield Units (HU) from Digital Imaging and Communications in Medicine (DICOM) data. Next, the image contrast is enhanced using a lung window setting (window width (WW) of 1500 HU and window level (WL) of −600 HU) to highlight lung structures. All voxel intensities outside of the upper and lower limits are assigned the value of the closest limit. Following this, nearest-neighbor interpolation is applied to obtain isotropic spatial resolution in the axial plane so that each pixel has a size of $1 \times 1$ mm$^2$. After spatial normalization, an image with standard bone window settings (WW: 1800, WL: 400) is saved, as it is used as an input in the lung isolation step of the workflow. In order to smooth the effect of different reconstruction methods on the image and to reduce the computational burden, intensity values are aggregated into bins of equal width. This also allows optimization of storage and image processing by packing the images into a much shorter 8-bit integer range and by filtering high-frequency noise. Hereafter, the image is cropped or padded with air intensity values to arrive at a resolution of $512 \times 512$ pixels, which is chosen as input for the selected deep learning architecture. All image processing and deep learning modeling steps were performed in Python 3.7 with the libraries and respective versions detailed in supplementary materials Table suppl. 1.

**Lung region isolation**. A robust algorithm for the isolation of the lung region was developed in order to focus on the ROI and allow for the use of whole-body CT scans as input. First, the CT couch is detected and removed from the image volume. Air-filled connected volumes are detected and region growing and morphological operations are applied in order to remove small vessels and to connect adjacent regions, resulting in a 3D binary lung mask. The spine axis is identified and the lung mask is halved and symmetrically flipped about the sagittal plane, keeping the union of the flipped and the original lung masks. By doing so, the algorithm is optimized for handling lung abnormalities such as atelectasis, pulmonary infiltration, consolidation, and fibrosis. To accurately identify the spine axis, a further algorithm was developed which identifies the center of the spine using the stored preprocessed image with bone window settings as described in the previous section (Fig. 14a suppl.). A "bone image" slice containing the lung is projected onto the coronal plane and filtered with a seventh-order moving average filter (Fig. 14b, c suppl.). This is repeated for the first five slices in which the lung mask is present in order to find a starting point for the center spine position $S_0$. The axis of the spine is positioned normally to this point (Fig. 14d suppl.).

$$S_0 = \frac{1}{n} \sum_{z=0}^{n} P_z \tag{1}$$

Where P is a central spine point for the current axial slice, $n$ is the number of slices (=5).

Due to irregularities in patient positioning and anatomy, the central spine position $S_t$ is recalculated slice-wise by using exponential smoothing:

$$S_t = \alpha \cdot x_t + (1 - \alpha) \cdot S_{t-1} \tag{2}$$

Where $x$ is a central spine point based on the filtered signal for the current axial slice, and α is the weighting coefficient (=0.3).

This method of flipping the lung mask allows for the inclusion of regions that contain large-sized abnormalities, such as lung collapse, which obscure parts of the lung, whereas commonly used methods exclude those regions (Fig. 14f, g suppl.).

A morphological dilation with the circle kernel ($r = -5$) is applied to the resulting lung mask in order to have a margin around the lung area. The final binary lung mask is used to isolate the lung region within the original image by setting all the voxel values outside the mask to the normalized air value.

**Tumor detection and segmentation.** The widely used 2D U-net convolutional neural network (CNN) was employed for slice-wise tumor segmentation[25–28]. The axial projection was used to train the network due to the higher resolution of image representation in this plane. To improve segmentation performance, several changes were made to the original CNN architecture. First, rectified linear unit (ReLU) activations were replaced with Exponential Linear Unit (ELU) in order to alleviate the gradient vanishing problem and kick-start the training process[29]. Second, dropout layers with the dropout rate ($p = 0.5$) were introduced prior to the two last layers of the U-net encoder to prevent overfitting[30].

A 2D CNN architecture was chosen for several reasons: (1) by using a 2D input the training dataset can be increased by more than a factor of 60, as overall more than 60,000 unique slices were available in the training set; (2) due to calculation costs, most present deep 3D architectures could analyze only a subvolume of the medical image[31,32], or they require a dimensionality reduction using interpolation or other image processing methods. 2D architectures do not have this problem and can process CT scans in the original resolution; (3) our main goal was to develop a pipeline that can be used in a clinical setting, and a 2D architecture allows for significantly lower requirements for executing PC. Our software does not require GPUs and can run on a regular laptop (Intel Core i5, 2.5 GHz, 8 GB RAM).

In order to increase the robustness of the system to a wide range of imaging parameters, the training dataset was expanded using augmentation techniques with the following parameters: random rotation around the image center pixel in a range of 0–25 degrees with a probability of 60%, random horizontal and vertical shifts of the image in the range of 12% of image shape with a probability of 25%, random zooming of the image with a maximum of 3% of the image shape with a probability of 10%.

The loss function was calculated by combining the Dice similarity coefficient (DSC) loss and the binary cross-entropy, and privilege was given to the DSC loss during the first 50 epochs. The privilege was defined by the coefficients before the DSC and cross-entropy terms in the loss function. By adding the binary cross-entropy component to the loss function, negative samples (slices without contour) could also contribute to the training.

The model was trained for 300 epochs using eight NVIDIA GTX 1080 Ti GPUs. The Adam algorithm was used for the stochastic optimization of the loss function[33]. The cosine annealing scheduler was used to adjust the learning rate during the training process. A checkpoint function tracking the DSC on the test dataset was used to keep the best weights.

Predicted 2D binary masks are stacked into a 3D volume and connected component extraction is applied as a post-processing step, whereby only spatially connected mask regions are extracted[34]. The connected region containing the most voxels is defined as the primary gross tumor target volume (GTV-1) for quantitative assessment. The final mask is resampled to the original image shape using cv2.INTER_BITS interpolation.

**Evaluation metrics.** In order to evaluate tumor detection performance, we generated lung-based labels, where lungs containing a tumor segmentation were assigned a positive label and lungs without were labeled negative. For cases where a tumor was present in both lungs of a patient, both were labeled positive. The ability of the system to detect tumors was assessed by calculating the area under the receiver operating characteristic curve (ROC AUC) and generating a confusion matrix.

Automatically generated binary masks were resampled to the original image resolution using cv2.INTER_BITS interpolation before comparing with manual segmentations. The contouring performance of the proposed pipeline, as well as the doctor's variability, were assessed by using the volumetric Dice similarity coefficient (DSC), Jaccard index (Ji), and 95th percentile Hausdorff distance (H95th). Additionally, we have evaluated quantitative contouring performance using Surface DSC and Added Path Length (APL).

The DSC is a measure of overlap between two volumes and was computed as:

$$\text{DSC}(A, B) = \frac{2 \cdot |A \cap B|}{|A| + |B|} = \frac{2 \cdot \text{TP}}{2 \cdot \text{TP} + \text{FP} + \text{FN}} \quad (3)$$

Jaccard index, used for gauging the similarity between two volumes, was computed as:

$$\text{Ji}(A, B) = \frac{|A \cap B|}{|A \cup B|} = \frac{\text{TP}}{\text{TP} + \text{FP} + \text{FN}} \quad (4)$$

where A and B are the sets of voxels corresponding to the ground truth and the automatic segmentation, respectively. TP is the number of true positive voxels, FP

is the number of false-positive voxels and FN is the number of false-negative voxels.

To evaluate the maximum deviation between the automatically segmented surface boundary and the ground truth surface boundary, the 95th percentile of Hausdorff distance (H95th) was used. Hausdorff distance (H) is defined as:

$$H(A, B) = \max\{\sup_{a \in Sa} \inf_{b \in Sb} d(a, b), \sup_{b \in Sb} \inf_{a \in Sa} d(b, a)\} \quad (5)$$

where a and b are the points on the voxel sets A and B, which represent the ground truth and the automatic segmentation, respectively. Sa and Sb are the surfaces of A and B.

Surface DSC at tolerance τ was computed as:

$$\text{SurfDSC}(A, B, \tau) = \frac{|S_a \cap \beta_b^{(\tau)}| + |S_b \cap \beta_a^{(\tau)}|}{|S_a| + |S_b|} \quad (6)$$

Where Sa and Sb are the surfaces of A and B, $\beta_a^\tau$ and $\beta_b^\tau$ are the border regions of A and B at a given tolerance τ, where τ is a maximum deviation from the ground truth contour which would not be penalized[35]. Tolerance τ for the NSCLC segmentation task have been evaluated on dataset 8 using segmentations of five experts.

APL was defined as follows:

$$\text{APL}\left(A, B, \text{PS}_{xy}\right) = 10 * \text{PS}_{xy} \sum B - A \cap B \quad (7)$$

Where A and B are the voxel sets of automatic and manual segmentation respectively and $\text{PS}_{xy}$ is the pixel spacing in the axial plane in mm[36].

In addition to the model performance evaluation on the test and validation datasets, the variability between expert clinicians was assessed and displayed against the performance of our method by comparing the volumetric DSC among all possible comparison pairs, i.e., experts were compared with each other as well as with the proposed method.

To better gauge the performance of our model under varying circumstances, it was evaluated with regard to slice-thickness, tumor complexity, tumor size, and tumor location. Tumor size subgroups were chosen based on the overall tumor size distribution in the training set. Furthermore, expert subjective tumor complexity labels were defined. To describe the complexity of the tumor, two medical doctors were asked to label the test and validation dataset as follows: for tumors where segmentation cannot be performed without a corresponding PET scan the labels were set to "1", and "0" otherwise. In case of disagreement, the label "1" was chosen. Additionally, one medical doctor have also labeled the tumor locations on the test and validation datasets, where tumor locations were defined as follows: lung parenchyma, mediastinum, and chest-wall involvement. Tumor locations were selected based on the discussion with clinical experts and existing published research[37].

**Statistical analysis.** For all non-normally distributed scores the median and interquartile range (IQR) were reported, as well as the frequency histograms[38]. Statistical significance was assessed using a two-sided Mann–Whitney–Wilcoxon test with Bonferroni correction. Survival evaluation was done in R (version 4.0.2) using survival (version 3.1–12) and survminer (version 0.4.7) packages. To estimate the difference between survival groups a log-rank test was applied. High and low survival groups were separated by the median tumor volume or median RECIST measurement respectively. Random sampling with a replacement bootstrapping strategy was used to compute confidence intervals for AUC values.

**An in silico clinical trial.** This trial was registered at clinicaltrials.gov (NCT04164186). For the first and second endpoints (the time needed for the processes of manual and automated segmentation, and inter and intra-observer variability), participants used a state-of the-art commercial software (MIM version 7.0.4) to produce the segmentations. In order to make the conditions of the trial close to the real clinical practice, experts had CT and PET scans available for each patient and they were able to use a semi-automated segmentation solution provided by MIM, while the proposed method generated the segmentation using only CT scans.

For the third endpoint (preference of experts for manual or automatically generated segmentations), a software tool was developed in-house. The tool has two interactive screens with the first screen showing the description of the experiment and a small questionnaire. In order to analyze preferences at different levels of expertize, the participants were asked to specify their training (e.g., radiologist, radiation-oncologist, medical doctor). The second screen displays comparisons between pairs of segmented axial CT slices (automatic vs. expert) with randomized screen positions, blinded to the participant. For each comparison pair, the participants were asked to select the more accurate contour. Finally, a table was generated containing the choices made. Screenshots of this tool are provided in supplementary materials (Figs. 15, 16 suppl.).

The software tool presents scans and contours from the external validation datasets 8. It randomly selects 100 pairs of contoured CT slices, where the DSC between the contours was higher than 0.7. During the assessment, participants were able to adjust the image contrast by changing window settings (WW and WL) and leaving comments.

The preference of the experts was evaluated using the qualitative preference score, defined as:

$$PS = \frac{n_m}{n_o} \times 100\%, \qquad (8)$$

where $n_m$ is the number of times where preference was given to the proposed method,

$n_o$ is a number of cases in total.

**Reporting Summary**. Further information on research design is available in the Nature Research Reporting Summary linked to this article.

## Data availability

The datasets 1, 6, 7, and 8 used in this study are available open-source and can be accessed through the corresponding sources: dataset 1—https://wiki. cancerimagingarchive.net/display/Public/NSCLC-Radiomics; dataset 6—https://wiki. cancerimagingarchive.net/display/Public/NSCLC+ Radiogenomics#28672347a99a795ff4454409862a398ffc076b98; dataset 7—https://wiki. cancerimagingarchive.net/display/Public/NSCLC-Radiomics-Genomics# 16056856db10d39adf704eefa 53e41edcf5ef41c; dataset 8— https://wiki. cancerimagingarchive.net/display/Public/NSCLC-Radiomics-Interobserver1# 52756590171ba531fc374829b21d3647e95f532c. The processed datasets 2, 3, 4, 5, 9, and 10 are available under restricted access as they were provided under Data Transfer Agreements from corresponding centers, and are not yet public due to data privacy laws, access can be obtained through the corresponding author upon request subject to ethical review. The approximate time for processing the data request is 1 month. The raw datasets 2, 3, 4, 5, 9, and 10 are protected and are not available due to data privacy laws. The minimum dataset is available on the GitHub repository of this project: https:// github.com/primakov/DuneAI-Automated-detection-and-segmentation-of-non-small-cell-lung-cancer-computed-tomography-images/tree/main/Software%20for% 20qualitative%20assesment/test_data. Philippe Lambin should be addressed for correspondence and material requests (email: philippe.lambin@maastrichtuniversity.nl)

## Code availability

Code, model files, extra software used in this manuscript, and derived data to reproduce the results are available on the GitHub page: https://github.com/primakov/DuneAI-Automated-detection-and-segmentation-of-non-small-cell-lung-cancer-computed-tomography-images. Code for the conversion of DICOM to NRRD format is available through Precision medicine toolbox[39] GitHub page: https://github.com/primakov/precision-medicine-toolbox

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

## Acknowledgements

S.P.P., M.B., and I.H. acknowledge the financial support of the Marie Skłodowska-Curie grant (PREDICT - ITN - No. 766276). A.I. acknowledges the financial support from the Liege-Maastricht imaging valley grant. P.L. and H.C.W. acknowledge financial support from ERC advanced grant (ERC-ADG-2015 n° 694812 - Hypoximmuno), ERC-2018-PoC: 813200-CL-IO, ERC-2020-PoC: 957565-AUTO.DISTINCT, SME Phase 2 (RAIL n°673780), EUROSTARS (DART, DECIDE, COMPACT-12053), the European Union's Horizon 2020 research and innovation program under grant agreement: ImmunoSABR n° 733008, FETOPEN- SCANnTREAT n° 899549, CHAIMELEON n° 952172, EuCanImage n° 952103, TRANSCAN Joint Transnational Call 2016 (JTC2016 CLEARLY n° UM 2017-8295), and Interreg V-A Euregio Meuse-Rhine (EURADIOMICS n° EMR4).

## Author contributions

S.P.P., A. J., A. I., and P.L. conceived the idea of the article. M.B., S.A.K., E.K., A.I., S.S., I. H., J.W., R.M., H.A.G., L.E.L. H., O.M., M.S., R.G., G.W., A.L., E.L., and X.G. participated in the data acquisition and clinical trial. S.P.P. implemented the analysis. E.L. and S.K. reproduced the results. J.E.v.T., A.J., A.I., H.C.W., P.L., S.A.K., and R.G. contributed to the writing of the manuscript. H.C.W, A.J., and P.L. supervised the work. P.L. approved the submitted version and has agreed both be personally accountable for the author's own contributions and to ensure that questions related to the accuracy or integrity of any part of the work, even ones in which the author was not personally involved, are appropriately investigated, resolved, and the resolution documented in the literature.

## Competing interests

S.P.P. reports, within the submitted work a non-issue, non-licensed patent in the field of medical imaging segmentation: IMAGE DATA PROCESSING METHOD, METHOD OF TRAINING A MACHINE LEARNING DATA PROCESSING MODEL, AND IMAGE PROCESSING SYSTEM; the year 2020; application number (PCT/NL2020/050794); inventors: S.P.P., H.C.W., P.L. reports, within and outside the submitted work, during the last 5 years, grants/sponsored research agreements from Varian medical, Oncoradiomics, ptTheragnostic, Health Innovation Ventures and Exomnis. He received an advisor/presenter fee and/or reimbursement of travel costs/external grant writing fee and/or in-kind manpower contribution from Oncoradiomics, BHV, Merck, Accuray, Elekta and Convert pharmaceuticals. Dr. Lambin has minority shares in the company Oncoradiomics SA, Comunicare Solutions SA, LivingMed Biotech, and Convert pharmaceuticals and is co-inventor of two issued patents with royalties on radiomics (METHOD AND SYSTEM FOR DETERMINING A PHENOTYPE OF A NEOPLASM IN A HUMAN OR ANIMAL BODY; the year 2014; publication number WO/2014/171830; IMAGE ANALYSIS METHOD SUPPORTING ILLNESS DEVELOPMENT PREDICTION FOR A NEOPLASM IN A HUMAN OR ANIMAL BODY; the year 2016; publication number WO/2016/060557) licensed to Oncoradiomics, one issue patent on mtDNA (METHOD FOR DETERMINING THE RISK OF DEVELOPING RADIATION-INDUCED TOXICITY AFTER EXPOSURE TO RADIATION; the year 2014; publication number WO/2014/184028) licensed to ptTheragnostic/DNAmito, three non-patentable inventions (software) licensed to ptTheragnostic/ DNAmito, Oncoradiomics, and Health Innovation Ventures. He confirms that none of the above entities or funding was involved in the preparation of this paper. Dr. Woodruff reports, outside of the current manuscript, (minority) shares in the company Oncoradiomics and non-issues, the non-licensed patent in the field of medical imaging segmentation. Dr. Lizza Hendriks reports, none related to the current manuscript, outside of current manuscript: research funding Roche Genentech, Boehringer Ingelheim, AstraZeneca (all institution); advisory board: Boehringer, BMS, Eli Lilly, Roche Genentech, Pfizer, Takeda, MSD, Boehringer Ingelheim, Amgen (all institution); speaker: MSD (institution); travel/ conference reimbursement: Roche Genentech (self); mentorship program with key opinion leaders: funded by AstraZeneca; fees for educational webinars: Quadia (self); interview sessions funded by Roche Genentech (institution); local PI of clinical trials: AstraZeneca, Novartis, BMS, MSD /Merck, GSK, Takeda, Blueprint Medicines, Roche Genentech, Janssen Pharmaceuticals, Mirati. The remaining authors declare no competing interests.
