## [Peer Review File · Nature Communications]

Reviewers' comments:

Reviewer #1 (Remarks to the Author): Expert in Digital pathology and segmentation

This is a well-written manuscript with extensive validation, including in-silico trial to assess usability of a deep learning automated lung tumor segmentation method. The paper is very easy to follow. Also nice is the comparison of the proposed method with a different deep learning method.

There are however several issues that need to be addressed:

1. Method novelty is rather limited. The proposed method uses a 2D Unet, which is not the state-of-the-art in deep learning. More importantly, a 2D-Unet has shown to be less accurate than several other methods including by the comparison method used. What is the rationale then to use a 2D-Unet? More importantly, are the accuracy gains due to 2DUnet or is it because of the pre-processing technique? In other words, could you use any technique with the pre-processing technique to get similar accuracy? It would have been more interesting to improve on the state-of-the-art with the pre-processing technique to see how much better you can get.

2. It is also some misleading to state that 1343 thoracic CT scans were analyzed. The testing itself was done on 93+236 patients. I think that needs to be spelled out more clearly.

3. Some of the stated novelties don't really warrant to be called method novelties. For example, image normalization, which is essentially intensity clipping is fairly common in all deep learning analysis methods -- its so common that it's sometimes not even mentioned to save space. So I suggest that this not be mentioned as a novelty. Unless of course, there is something unique about what is done here, which then needs to be explained better to clarify why this is just not an intensity clipping based normalization. Similar is the loss changing function, which is maybe incrementally novel as some methods even learn the weighting of the losses as part of the training.

4. By far, the more novel aspect of this work is the non-lung removal technique that leaves only the lung and removes heart and chestwall. These usually are confounders for segmentation and it seems like this technique helps to improve segmentation performance. However, no results are presented to show the impact of this technique -- that is results with and without it. Whereas the authors say that this technique is robust, no experiments are shown to clearly demonstrate how robust it is. From the description and the Figure 3 - this looks rather adhoc (Figure 3 looks like a fixed morphological expansion in some slices). The authors are strongly encouraged to expand on this technique and elaborate the effect of different imaging conditions on how well this pre-processing works and what happens when it fails etc.

5. Comparison to the published method: While the comparison using the patch as was (developed?) for that method looks OK, its not clear why the fully automated configuration is a good comparison. A fairer comparison would to be apply the pre-processing as used in this approach for that method. Also, were the patches intensity clipped for that method also. Ideally, the method needs to be trained under the same/similar conditions to provide a side-by-side comparison. Please expand on this aspect.

6. Results: Its not clear why slice-wise detection accuracy is relevant here? If you don't detect on a slice,

the volumetric accuracy would go down anyway. I think reporting the segmentation accuracy including the Hausdorff metrics is more relevant than the detection AUC. Also how was the detection flagged as detection? A single pixel detected or more than x% of the tumor pixels flagged correctly? This makes a difference and could lead to overly optimistic results depending on the criteria for detection.

Dice and Jaccard are highly correlated. Instead a better metric to report would be Hausdorff distance and Dice. The validation accuracy of 0.77 DSC is modest and is highly similar to other prior works. Is a DSC of 0.77 sufficiently accurate for clinical use? Of course, the authors show a better performance on the 22 dataset of 0.86 for the proposed and 0.83 for the compared method but those were 22 cases (and could potentially be larger tumors, which would have obviously biased towards a higher accuracy). This is particularly problematic given that the user preference of the algorithm was around 55%. Is this sufficient then? Also, while the user preference is good, a more quantitative metric such as the added path length as used by Vassen et. al <https://pubmed.ncbi.nlm.nih.gov/33458300/> is a better metric for judging the usefulness of an algorithm. In general, DSC is useful because several other papers use it to establish a common reference framework for comparison. But mm distance based metrics like Hausdorff distances are far more useful to understand why a particular algorithm can be helpful.

7. Figure 2: The number of patients in the PET needed for the larger tumors seem to be 5 cases. Also the error bars are huge for the validation set. In general, as the validation set is used for model selection, it's less critical than the testing set, which is completely blinded to the model selection and training. I suggest that you put the validation set results in supplements and focus more on the testing set.

8. The DSC results seem to use median in some place and average in others. Also why is IQR one number? Don't you report the lower 25% and the upper 75% values. For the survival analysis, did you also perform Cox regression using other confounders to see how useful the segmentation really is?

9. Discussion: Some of the statements are misleading -- First the novelty related harmonization, changing loss functions etc are not really novel. The use of CTs with lung abnormalities, negative examples etc is mentioned for the first time here. Infact, if these negative examples were important for improved accuracy, their use needs to be evaluated with ablation experiments.

Second, Is the comparison method Jue et.al or Jiang et.al? Also, the comparison of their test results of 0.68 to the one used here is highly misleading. Did they use the same testing dataset? It looks like they used the LIDC dataset, which has smaller lung nodules rather than large biopsy positive lung tumors treated with radiation therapy as used in this work. It's the same for other prior works and their results. In general, it's not a fair comparison of other results evaluated on entirely different test sets than what was used here. It would be better to rephrase and contextualize other related works better.

I think the two most interesting parts of this work were the in-silico trial (it could be made even better by using added path length and other quantitative metrics) and the comparison with and without PET results. It would be useful to focus the paper more towards these rather than on several other aspects like the KM analysis.

Reviewer #2 (Remarks to the Author): Digital medicine and prediction

This is a very interesting and well-designed study on building and evaluating a fully automatic pipeline for the detection and 3D segmentation of non-small cell lung cancer on CT. Data from 8 institutions and 1328 thoracic CT scans were analyzed. Performance of the automated pipeline was compared to radiologist and radiation oncologist contours. Additionally, RECIST and volume measurements were derived from the automatic segmentations to assess the prognostic power to stratify patients into high and low risk survival groups compared to manual segmentations. Overall, this is an extremely well designed and well written study. The comparisons performed in the “in silico” clinical trial are very valuable, demonstrating the utility of this tool and potential for clinical translation. Overall, this is a very novel study and I believe this tool will be very useful in many clinical applications and future research studies. I only have minor suggestions.

1. The unmet need this tool addresses is well described, and this tool has the potential to improve the clinical workflow in both radiotherapy and radiology. One of the applications listed is on automated screening, however this isn't discussed in the introduction prior to this point. It may be helpful to add a few sentences on the role of automated detection and segmentation in the screening setting, and I have the same comment regarding adaptive re-planning.
2. Figure 3 is a bit hard to visualize. Consider a different layout avoiding the overlapping regions to improve readability, or focus only on the zoomed in regions to allow for better visualization of the segmentation borders.
3. Figure 5: For a direct comparison, it would be beneficial to have the Dr's on the x axis in the same order in both figures.
4. Figure 6: The x-axis is a bit hard to read with respect to the comparisons.
5. Results: Based on DSC results, it looks as though some tumors were not segmented accurately. It would be valuable to provide a summary of these results on how many tumors were unable to be segmented or resulted in inaccurate contours (that may have needed editing). Similarly, a description of these cases in the discussion would be valuable.
6. Discussion: The authors stated in the start of the discussion that all patients had cancer, therefore they weren't able to evaluate patient based AUC for lung cancer detection. However it then state that “positive impact on the management of lung cancer patients, as it will improve the detection accuracy.” This needs to be removed or re-worded as it is not supported by the results.
7. Methods: Please provide the final number of training images used to train the CNN in the methods.

Reviewer #3 (Remarks to the Author): Lung cancer imaging

This paper addresses an automated delineation tool for primary lung cancer lesions. The authors developed their own algorithm and applied this to: 1. Diagnosis 2. Delineation for radiotherapy planning purpose 3. A clinical dataset for the relation with survival. The authors used 10 datasets from 8 institutions.

I would like to compliment the authors for their work and I believe that the delineation tool is promising. However, I do have major concerns with respect to the data and manuscript in general.

1. The patient characteristics were not described in the manuscript. What was the disease stage of the patients? Patients with nodal and / or distant metastases? Were the patients treatment naïve?

2. The manuscript reads as two papers in one. The same delineation tool was used, but two different applications were described; diagnosis and radiotherapy planning. For the first dataset I would use patients without a definitive diagnosis (e.g. a lung cancer screening cohort), while a different cohort should be used for radiotherapy planning (patients with different T stages and a pathologically proven diagnosis of NSCLC). However, it remains unclear what patients were enrolled and it seems that the same cohort of patients was used to answer both questions.

3. In the results section, the authors compare survival for the automated delineation tool and manual RECIST measurements. The median volume was used to discriminate between two groups per method. However, without any knowledge of patient characteristics and nodal stage and disease / treatment history, it seems inappropriate to perform such an analysis.

I would suggest the authors to rewrite the manuscript and focus on automated delineation for radiotherapy planning. In a different manuscript the tool can be evaluated on a separate lung cancer screening cohort to evaluate the diagnostic value.

The prognostic value is not a very interesting topic in my opinion. However, what could be interesting is response evaluation (automated delineation vs manual).

Replies to Reviewers' comments

We thank the reviewers for the comments and questions, although we disagree with several statements. Where we are in agreement we have modified the Results/Methods sections of the article, including new metrics (SurfaceDSC, APL) and updated quantitative performance.

Reviewer #1 (Remarks to the Author): Expert in Digital pathology and segmentation

This is a well-written manuscript **with extensive validation**, including in-silico trial to assess usability of a deep learning automated lung tumor segmentation method. The paper is very easy to follow. Also nice is the comparison of the proposed method with a different deep learning method.

Answer: We appreciate the reviewer's complementary evaluation. We are aware that Nature Communications has a broad readership therefore we appreciate the comment on the ease of reading the paper.

There are however several issues that need to be addressed:

1. Method novelty is rather limited. The proposed method uses a 2D Unet, which is not the state-of-the-art in deep learning. More importantly, a 2D-Unet has shown to be less accurate than several other methods including by the comparison method used. What is the rationale then to use a 2D-Unet? More importantly, are the accuracy gains due to 2DUnet or is it because of the pre-processing technique? In other words, could you use any technique with the pre-processing technique to get similar accuracy? It would have been more interesting to improve on the state-of-the-art with the pre-processing technique to see how much better you can get.

Answer: We respectfully disagree with the statement that the novelty is limited. In this work, we present an automatic segmentation method as a whole with several novel steps introduced allowing for fully automatic segmentation of CT scans and not the DL architecture alone. Detailed description of particular steps' novelty can be found in discussion (lines 304-315). The novelty aspect is also demonstrated by a submitted patent (PCT/NL2020/050794), which has received a very positive search report.

The accepted method of measuring solo DL architecture performance is applying it to a dataset reporting set of metrics, and is very problem-specific from task to task. Moreover, it is highly dependent on the pre-processing of the input data. Whereas several studies showed that 3D architectures can reach a slightly higher performance in terms of DSC for their particular problem ¹, 3D architectures require quite some time to operate and computational power to run, whereas the 2D architectures are faster and as we show in this work in the combination with an advanced pre-processing and 3D post processing can outperform 3D architectures. We believe that pre-processing of the CT scan can have a higher impact on the pipeline performance than the difference in the model architecture, therefore choice of a DL architecture should be problem specific and should not be based only on one metric performance. Whether or not complex 3D architectures are feasible for Kaggle competitions,

they are not always the best choice for real clinical applications. We present an algorithm for fully automatic segmentation of CT scans which can be used by doctors/researchers/students on most regular laptops without a GPU and can run on most of the popular operating systems (Windows, MacOS, Linux) supporting python 3.7. We also would like to emphasize the novelty of the extensive validation: comparison with several experts, registered In Silico Trial, use of 5 quantitative metrics for segmentation performance and assessment of prognostic power of segmentations (a metrics requested by the clinicians).

2. It is also some misleading to state that 1343 thoracic CT scans were analyzed. The testing itself was done on 93+236 patients. I think that needs to be spelled out more clearly.

Answer: We politely disagree with a reviewer. We do not use the misleading statement “1343 thoracic CT scans were analyzed” anywhere in the article. We do not state that 1343 CT scans were used for testing anywhere in the article. Moreover the article has a Table 1 (line 161), which shows exactly how each dataset was used. Below are the lines where we mention a total number of scans:

Lines 49-51: “*We developed and validated a fully automated pipeline for the detection and volumetric segmentation of non-small cell lung cancer (NSCLC) using 1343 thoracic CT scans from 8 institutions.*”

Lines 123-125: “*Overall, 1328 thoracic volumetric CT scans with corresponding 3-dimensional tumor segmentations were used in order to train, test and validate a fully automated method for detection and segmentation of NSCLC in standard-of-care images*”

Lines 376-377: “*The CT scans of 1343 NSCLC patients were retrospectively collected and anonymized by each center and approved by the respective institutional review boards.*”

3. Some of the stated novelties don't really warrant to be called method novelties. For example, image normalization, which is essentially intensity clipping is fairly common in all deep learning analysis methods -- its so common that it's sometimes not even mentioned to save space. So I suggest that this not be mentioned as a novelty. Unless of course, there is something unique about what is done here, which then needs to be explained better to clarify why this is just not an intensity clipping based normalization. Similar is the loss changing function, which is maybe incrementally novel as some methods even learn the weighting of the losses as part of the training.

Answer: We don't seem to understand this comment as we do not list “image normalization” in the novelties. If we correctly understand the reviewer refers to lines 306-307: “*1) a harmonization routine for the pre-processing of CT scans in order to more comprehensively unify patterns on the images for the models to learn from;*”. As this is not just an “*intensity clipping*”, we describe the harmonization routine in detail in the lines 388-406, which is also part of our patent (PCT/NL2020/050794).

4. By far, the more novel aspect of this work is the non-lung removal technique that leaves only the lung and removes heart and chestwall. These usually are confounders for segmentation and it seems like this technique helps to improve segmentation performance. However, no results are presented to show the impact of this technique -- that is results with and without it. Whereas the authors say that this technique is robust, no experiments are shown to clearly demonstrate how robust it is. From the description and the Figure 3 - this looks rather adhoc (Figure 3 looks like a fixed morphological expansion in some slices). The authors are strongly encouraged to expand on this technique and elaborate the effect of different imaging conditions on how well this pre-processing works and what happens when it fails etc.

Answer: We appreciate the reviewer's complementary evaluation on the non-lung removal technique which simplifies the process of segmentation.

The goal of this article was to develop a fully automated NSCLC segmentation method and extensively evaluate the performance of the method as a whole. Method design and impact of each particular step on the performance of the method was evaluated during the method development on the test set and is not reported as it is outside of the scope of this study. Our lung isolation method is computer vision based, has been evaluated by clinicians and it worked for all 1328 CT scans with a variety of different parameters, we believe that makes it robust.

5. Comparison to the published method: While the comparison using the patch as was (developed?) for that method looks OK, its not clear why the fully automated configuration is a good comparison. A fairer comparison would to be apply the pre-processing as used in this approach for that method. Also, were the patches intensity clipped for that method also. Ideally, the method needs to be trained under the same/similar conditions to provide a side-by-side comparison. Please expand on this aspect.

Answer: We provide comparisons in two settings since our method is fully automatic and we wanted to compare the methods under equal conditions regarding automation. It also highlights how much more complex the fully automatic segmentation is compared to a patch based approach and demonstrates that our proposed method, while performing a more complex task, can outperform a 3D architecture performing a seemingly easier task. We report on both settings because we felt this is the fairest comparison.

Furthermore, the point of the comparison was to compare different methods end-to-end, not the underlying architectures or pre-processing steps in particular. We tried to take a clinically relevant approach to the comparison. The external method uses its own pre-processing, for benchmarking the external model we used the code provided by authors without any modifications.

6. Results: Its not clear why slice-wise detection accuracy is relevant here? If you don't detect on a slice, the volumetric accuracy would go down anyway. I think reporting the segmentation accuracy including the Hausdorff metrics is more relevant than the detection AUC. Also how was the detection flagged as detection? A single pixel detected or more than x% of the tumor pixels flagged correctly? This makes a difference and could lead to overly optimistic results depending on the criteria for detection.

Dice and Jaccard are highly correlated. Instead a better metric to report would be Hausdorff distance and Dice. The validation accuracy of 0.77 DSC is modest and is highly similar to other prior works. Is a DSC of 0.77 sufficiently accurate for clinical use? Of course, the authors show a better performance on the 22 dataset of 0.86 for the proposed and 0.83 for the compared method but those were 22 cases (and could potentially be larger tumors, which would have obviously biased towards a higher accuracy). This is particularly problematic given that the user preference of the algorithm was around 55%. Is this sufficient then? Also, while the user preference is good, a more quantitative metric such as the added path length as used by Vassen et. al <https://pubmed.ncbi.nlm.nih.gov/33458300/> is a better metric for judging the usefulness of an algorithm. In general, DSC is useful because several other papers use it to establish a common reference framework for comparison. But mm distance based metrics like Hausdorff distances are far more useful to understand why a particular algorithm can be helpful.

Answer: We thank the reviewer for the suggestion of using the Added Path Length which we have implemented as well as Surface Dice Coefficient. Moreover, we have established the tau coefficient allowing for use of Surface Dice Coefficient for other studies performing NSCLC automatic segmentation.

Based on this comment we decided to drop the slice-wise accuracy evaluation. Since it is hard to find healthy lung CT's appropriate for this study, we have evaluated the NSCLC detection per lung, where labels were assigned as follows: lungs containing a tumor segmentation had a positive label and those without were labeled negative. For cases where a tumor presented in both lungs both were labeled positive. More details can be found in the Methods, Evaluation metrics section.

We do not understand the reviewer's concern about the Hausdorff distance, as we did report Dice, Jaccard, and Hausdorff distance in our work for both the test and the external validation. The reason for reporting Jaccard is to increase the comparability to other works. However, we have added the Added Path Length and Surface Dice for both test and external validation after the reviewer's suggestion, which would make our article even more open for comparison with other work.

To our knowledge there are no articles which use such an extensive external validation dataset of 236 patients and show robust higher segmentation performance on both test and external validation, using only the whole CT scan (and not cropped regions, or PET). In this work, we have extensively evaluated the performance of our method not only quantitatively but also introducing a novel framework for evaluating a qualitative performance in a prospective manner. In short in the new version of the paper, contains now 1) Dice, 2) Jaccard, 3) Hausdorff distance, 4) the Added Path Length and 5) Surface Dice for both test and external validation, next to the preference of the users and the correlation with survival.

Concerning the comments: The validation accuracy of 0.77 DSC is modest and is highly similar to other prior works. Is a DSC of 0.77 sufficiently accurate for clinical use? Of course, the authors show a better performance on the 22 dataset of 0.86 for the proposed and

0.83 for the compared method but those were 22 cases (and could potentially be larger tumors, which would have obviously biased towards a higher accuracy). This is particularly problematic given that the user preference of the algorithm was around 55%. Is this sufficient then?

Answer: We thank the reviewer for this comment, we have reevaluated the way we were calculating the performance. Since most of our datasets have only GTV1 contoured, DSC was lower for our model because it was detecting all the cancerous regions on CT. We have added a post-processing step for a quantitative evaluation where we would leave only the biggest connected region, which has increased model performance in terms of all quantitative metrics. The current model performance in terms of DSC on the test is 0.85 and validation is 0.82. Additionally, we want to point out that DSC is based on manual segmentation which is highly unreliable (see inter Dr variability of [0.81-0.84] DSC). This is why after discussion with clinicians we have used several endpoints that are seen as more relevant from the prospects of the future users:

- In Silico trials (preregistered in www.clinicaltrials.gov: number: xx)
 - Time
 - Preference
 - Repeatability of Dr and software
- Correlation with survival
 - Dataset 1
 - Dataset 2

Even if the accuracy of the tools or the preference for the automated segmentation is comparable manual ones, the AI tools are second to none in terms of reproducibility and time needed to execute the task. Both the preference endpoint and the survival curve point out that the automated tools are not inferior and perhaps superior to the manual approach.

7. Figure 2: The number of patients in the PET needed for the larger tumors seem to be 5 cases. Also the error bars are huge for the validation set. In general, as the validation set is used for model selection, it's less critical than the testing set, which is completely blinded to the model selection and training. I suggest that you put the validation set results in supplements and focus more on the testing set.

Answer: We agree that it is important to focus on the independent datasets not used for training. In our group, we follow, like many others, the terminology where the test dataset is a dataset used for fine tuning the model and external validation set is used for external validation of the model. We describe these steps in the lines 125-128: “*Datasets 1-7 were combined and randomly divided into training and testing datasets with 999 patients and 93 patients, respectively (see Table 1). Datasets 8-10, comprising 236 patients were used for external validation of the method. A summary of the data is provided in Table 1.*” Datasets 9-10 were completely blinded to the model selection and training and were used only as an external validation. We have added systematically the term “external” to validation to emphasize that the dataset was blinded for the model.

8. The DSC results seem to use median in some place and average in others. Also why is IQR one number? Don't you report the lower 25% and the upper 75% values. For the survival analysis, did you also perform Cox regression using other confounders to see how useful the segmentation really is?

Answer: The reporting of average value is based on the statistics. If the distribution is far from normal we use Median and IQR. We do not report mean DSC anywhere in the article since it is not distributed normally. We report mean time, since time is normally distributed in this work.

Regarding IQR, we use the following definition: *“In descriptive statistics, the interquartile range, also called the midspread, middle 50%, or H- spread, is a measure of statistical dispersion, being equal to the difference between 75th and 25th percentiles, or between upper and lower quartiles, $IQR = Q_3 - Q_1$.”*

We did not perform cox regression at all. We used the median to split the patients in high and low survival groups. Performing cox regression with other confounders is outside of the scope of this study and can be a separate article itself. We see the correlation with survival as another metric to validate the manual and automated segmentations. There are several metrics to evaluate automatic segmentation. The challenge is that the gold standard, the manual segmentation, has a large intra Dr variability. So we propose here a new approach which has a strong clinical flavor. Our hypothesis is that if the manual segmentation, which is the gold standard in the clinic, would have been better then the automated segmentation we would have expected a better correlation (or a larger split) between survival and the metrics returned from the automated segmentation compared to the manual one. We see the reverse. This reassures us that our method is at least as good as current gold standard. We have had several discussions with potential users (radiologists, radiation oncologists) and we notice that this result is more compelling for them than the DICE index, Jaccard, or Hausdorff distance. In other words, we have tried to acknowledge the different prospects of data scientists and clinicians on this matter.

9. Discussion: Some of the statements are misleading -- First the novelty related harmonization, changing loss functions etc are not really novel. The use of CTs with lung abnormalities, negative examples etc is mentioned for the first time here. Infact, if these negative examples were important for improved accuracy, their use needs to be evaluated with ablation experiments.

Second, Is the comparison method Jue et.al or Jiang et.al? Also, the comparison of their test results of 0.68 to the one used here is highly misleading. Did they use the same testing dataset? It looks like they used the LIDC dataset, which has smaller lung nodules rather than large biopsy positive lung tumors treated with radiation therapy as used in this work. It's the same for other prior works and their results. In general, it's not a fair comparison of other results evaluated on entirely different test sets than what was used here. It would be better to rephrase and contextualize other related works better.

I think the two most interesting parts of this work were the in-silico trial (it could be made even better by using added path length and other quantitative metrics) and the comparison

with and without PET results. It would be useful to focus the paper more towards these rather than on several other aspects like the KM analysis.

Answer: We politely disagree with the reviewer on this question. Reviewer's statement about novelty is a repetition of comment 3, which we have already commented on. The impact of every step in the pre-processing routine was tested on the test set (fine tuning dataset) during model fine-tuning. In fact, there were several steps such as noise reduction and kernel harmonization routine, which were dropped because they did not improve the performance on the test set. These steps are considered a method development and only the final configuration of the method, which showed the best performance, is reported, analogous to reporting DL architectures. In this article, we focused on reporting the method as a whole and performance of the whole method.

From the article, lines 208-209: “A previously published external segmentation model 19 was evaluated on dataset 8 and compared to our model.”

“19. Jiang, J. et al. Multiple Resolution Residually Connected Feature Streams for Automatic Lung Tumor Segmentation From CT Images. *IEEE Trans. Med. Imaging* 38, 134–144 (2019)”

To set our model in the context of similar published work we reported the results of previous published work, we did not directly compare. Authors of prior published studies report only averaged performance, which makes the comparison harder. From our side we report performance with regards to slice thickness, tumor complexity label and tumor size with the intention that it will make comparisons with other studies more clear.

Reviewer #2 (Remarks to the Author): Digital medicine and prediction

This is a very interesting and well-designed study on building and evaluating a fully automatic pipeline for the detection and 3D segmentation of non-small cell lung cancer on CT. Data from 8 institutions and 1328 thoracic CT scans were analyzed. Performance of the automated pipeline was compared to radiologist and radiation oncologist contours. Additionally, RECIST and volume measurements were derived from the automatic segmentations to assess the prognostic power to stratify patients into high and low risk survival groups compared to manual segmentations. Overall, this is an extremely well designed and well written study. The comparisons performed in the “in-silico” clinical trial are very valuable, demonstrating the utility of this tool and potential for clinical translation. Overall, this is a very novel study and I believe this tool will be very useful in many clinical applications and future research studies. I only have minor suggestions.

Answer: We thank the reviewer for his positive comments.

1. The unmet need this tool addresses is well described, and this tool has the potential to improve the clinical workflow in both radiotherapy and radiology. One of the applications listed is on automated screening, however this isn't discussed in the introduction prior to this point. It may be helpful to add a few sentences on the role of automated detection and segmentation in the screening setting, and I have the same comment regarding adaptive re-planning.

Answer: We thank the reviewer for the comment. We agree with the reviewer. We have added the following to the introduction:

Page 3, lines 71-76: “*Accurate segmentations of the tumor and organs at risk are also essential as errors might lead to over- or under-irradiation of both the tumor and/or healthy tissue. It has been estimated that a 1mm shift of the tumor segmentation could affect the radiotherapeutic dose calculations by up to 15%^{2,3}. Therefore, automated accurate segmentations can significantly reduce the time needed by clinicians to plan treatment, and adapt further treatment depending on the changes in the tumor.*”

Pages 3-4, lines 101-105: “*Automated segmentation of NSCLC tumors requires prior identification of the lesion as NSCLC. Invasive tissue biopsy is currently the clinical gold standard in identifying NSCLC. However, an accurate automated segmentation tool requires high detection accuracy. Therefore, a software that can automatically segment NSCLC tumors could also be used as a detection method, decreasing the need for invasive biopsies.*”

2. Figure 3 is a bit hard to visualize. Consider a different layout avoiding the overlapping regions to improve readability, or focus only on the zoomed in regions to allow for better visualization of the segmentation borders.

Answer: We thank the reviewer for the comment, we have updated the figure accordingly.

3. Figure 5: For a direct comparison, it would be beneficial to have the Dr's on the x axis in the same order in both figures.

Answer: We thank the reviewer for the comment, we have updated the figure.

4. Figure 6: The x-axis is a bit hard to read with respect to the comparisons.

Answer: We thank the reviewer for the comment, we have increased the text size for the x axis of the figure.

5. Results: Based on DSC results, it looks as though some tumors were not segmented accurately. It would be valuable to provide a summary of these results on how many tumors were unable to be segmented or resulted in inaccurate contours (that may have needed editing). Similarly, a description of these cases in the discussion would be valuable.

Answer: We thank the reviewer for the comment, we have provided the figure showing the cases where segmentation failed in the supplementary materials (suppl. Figure 15). Additionally, using DSC it is possible to count the number of cases where tumor was not segmented (DSC is zero) or cases where the segmentation was not proper (DSC<0.6) based on the figures in the results. We have also provided the Table 3 where we show how segmentation performance in terms of DSC depends on the tumor and scan characteristics. Unfortunately, to provide the description of the amount of tumors that have to be edited after

automatic segmentation we would need to organize another clinical trial and enroll participants to evaluate this aspect, this was originally out of scope of this study. However, we provide the software along with the article so that it can be used by researchers and tested for such aspects.

6. Discussion: The authors stated in the start of the discussion that all patients had cancer, therefore they weren't able to evaluate patient based AUC for lung cancer detection. However it then state that "positive impact on the management of lung cancer patients, as it will improve the detection accuracy." This needs to be removed or re-worded as it is not supported by the results.

Answer: We thank the reviewer for the comment, we decided to drop the slice-wise detection and use lung-wise instead, where the lung labels were assigned as follows: lung with tumor segmentation had a positive label and lung without tumor was labeled negative. For cases where tumor presented in both lungs both were labeled positive. More details can be found in the Methods, Evaluation metrics section. We understand that it is an artificial measure but it can provide a better insight on the detection performance of NSCLC for our method.

7. Methods: Please provide the final number of training images used to train the CNN in the methods.

Answer: We thank the reviewer for the comment, we provide the number of slices which were used to train the model in table 1, line 162. Number of original CT slices used for training/testing: 64896. Number of original CT slices used for validation: 25666. Additionally, data augmentation technique have been used during training, increasing the number of training entries.

We thank the reviewer for the questions, we are very pleased to get such positive feedback on our work. We have modified the Results/Methods sections of the article and updated the figures mentioned by the reviewer.

Reviewer #3 (Remarks to the Author): Lung cancer imaging

This paper addresses an automated delineation tool for primary lung cancer lesions. The authors developed their own algorithm and applied this to: 1. Diagnosis 2. Delineation for radiotherapy planning purpose 3. A clinical dataset for the relation with survival. The authors used 10 datasets from 8 institutions.

I would like to compliment the authors for their work and I believe that the delineation tool is promising.

Answer: We thank the reviewer for his comments. We rephrase the introduction to clarify the objective which is to build one comprehensive, ubiquitous tool to detect and segment lung tumors. We do not want to limit this tool to specific applications (radiology, radiotherapy or response evaluation). The advanced patented pre-processing pipeline allows it to work without distinction with radiology or radiotherapy simulation CT.

We see the correlation with survival as another metrics to validate the goodness of the manual and automated segmentation. Validating the tool for a specific indication in a real

world environment and building a prognostic model is beyond the scope of this paper. There are several metrics to evaluate automatic segmentation. The challenge is that the gold standard, the manual segmentation, has a large intra Dr variability. Therefore, we propose here a new approach, which has a strong clinical flavor. Our hypothesis is that if the manual segmentation, which is the gold standard in our days, would have been better than the automated segmentation we would have expected a better correlation (or a larger split) with survival of the automated segmentation compared to the manual one. We see the reverse. This reassures us that our method is at least as good as the manual one. We have had several discussions with potential users (radiologists, radiation oncologists) and we notice that this result is more compelling for them than the DICE index, Jaccard and Hausdorff distance. In other words, we have tried to acknowledge the different prospects of data scientists and clinicians on this matter.

However, I do have major concerns with respect to the data and manuscript in general.

1. The patient characteristics were not described in the manuscript. What was the disease stage of the patients? Patients with nodal and / or distant metastases? Were the patients treatment naïve?

Answer: We thank the reviewer for the comment. We have added the population description in the supplement Table 2.

2. The manuscript reads as two papers in one. The same delineation tool was used, but two different applications were described; diagnosis and radiotherapy planning. For the first dataset I would use patients without a definitive diagnosis (e.g. a lung cancer screening cohort), while a different cohort should be used for radiotherapy planning (patients with different T stages and a pathologically proven diagnosis of NSCLC). However, it remains unclear what patients were enrolled and it seems that the same cohort of patients was used to answer both questions.

Answer: We thank the reviewer for the comment. The objective of the paper is to build and validate a comprehensive, ubiquitous tool to detect and segment lung tumors. These automatic segmentations can then be used for planning/adapting treatment. By default, automatic segmentation would first require detection of the lesion, and the treatment planning is dependent on the segmentation of the tumor. Therefore, we believe that the tool is universal, and its application is dependent on the clinical need investigated. We do not want to limit this tool to specific applications.

Regarding the use of the same tool for both tasks, we used the dataset differently to assess the performance of the algorithm with regards to the application. For the detection, we used both right and left lungs as different entries. If a lung contains a tumor, it is labeled positive, and the detection performance was calculated accordingly. For the segmentation task, we used the DSC score to evaluate the performance. Therefore, while the same tool is applied, different statistical approaches were applied. We also tried to include patients with a wide range of clinical characteristics to help the model identify NSCLC tumors in its all stages and across different populations. Our advanced patented pre-processing pipeline allow to work without distinction with radiology (diagnostic question, response evaluation) or radiotherapy simulation CT. The most obvious experiment to address the question of the three potential applications is indeed via dedicated prospective clinical trials in real world environment. This

is exactly what we're doing in the context of a recently approved grants. In this paper we want to describe the new tool and how we validated it with more than 7 metrics.

3. In the results section, the authors compare survival for the automated delineation tool and manual RECIST measurements. The median volume was used to discriminate between two groups per method. However, without any knowledge of patient characteristics and nodal stage and disease / treatment history, it seems inappropriate to perform such an analysis. I would suggest the authors to rewrite the manuscript and focus on automated delineation for radiotherapy planning. In a different manuscript the tool can be evaluated on a separate lung cancer screening cohort to evaluate the diagnostic value.

The prognostic value is not a very interesting topic in my opinion. However, what could be interesting is response evaluation (automated delineation vs manual).

Answer: We thank the reviewer for this comment, and agree that a prognostic factor is not interesting. As explained above, we use the survival curve as an extra validation metrics of the segmentation tool. The clinicians we collaborate with are not totally convinced by a DICE index, Jaccard and Hausdorff distance but they have praised our approach regarding survival curve splitting. We used Kaplan Meyer curves only to visualize the prognostic power of segmentations, we did not perform cox regression and in this work we do not report any prognostic models, therefore we did not look at other confounders, patients characteristic or tumour stage (which are the same for the two segmentation tools because we use the two segmentation on the same cohorts, each group is therefore its own control).

In summary, in this work we used the KM curves only to look at the comparison of manual and automatic segmentations from another angle on request of the future users.

We thank the reviewers again for their helpful comments. We have made a determined effort to address each of the points raised by the reviewers, we have added new metrics, improved figures and provided the population description (Supplementary table 2). We believe that the manuscript is now much stronger as a result, and we hope that the revised manuscript is now acceptable for publication.

References

1. Nemoto, T. *et al.* Efficacy evaluation of 2D, 3D U-Net semantic segmentation and atlas-

based segmentation of normal lungs excluding the trachea and main bronchi. *Journal of Radiation Research* vol. 61 257–264 (2020).

2. Barrett, A., Dobbs, J. & Roques, T. *Practical Radiotherapy Planning Fourth Edition*. (CRC Press, 2009).
3. Stroom, J. C. & Heijmen, B. J. M. Geometrical uncertainties, radiotherapy planning margins, and the ICRU-62 report. *Radiother. Oncol.* **64**, 75–83 (2002).

REVIEWER COMMENTS

Reviewer #1 (Remarks to the Author):

The authors have addressed a few concerns but did not address the major concerns including the method novelty and fair comparisons to other methods.

Authors' response that "3D architectures require quite some time to operate and computational power to run, whereas the 2D architectures are faster" is not true. 3D methods are just as fast as 2D methods. Several methods published in computer science conferences and journals (papers by Summers, Roth, Rueckert, Pheng-Ahn Heng, Kevin Zhou, etc) have shown feasibility of using 3D methods.

Similarly, the novelty of image harmonization is claimed but is not really explained in detail as to why this is novel. The method description is standard intensity clipping based normalization.

In summary, while there are some merits such as evaluation with respect to inter-raters, and use of multiple open-source datasets, method is very limited in novelty and the lack of fair comparisons to current state of the art methods, makes it difficult to assess the method's performance using fair settings.

Reviewer #2 (Remarks to the Author):

I would like to commend the authors for their detailed responses to each of the reviewer comments. I believe the changes have improved the readability of both the text and figures. Congratulations on this very interesting, well designed, and easy to follow study!

Reviewer #3 (Remarks to the Author):

The authors addressed the questions and remarks well. I believe that the revised manuscript is of sufficient quality to be published in the journal.

Reviewer #4 (Remarks to the Author):

This study presents a fully automated deep learning pipeline for the detection and volumetric segmentation of non-small cell lung cancer (NSCLC). Major strengths of the study include: use of a large multi-institution dataset of 1328 thoracic CT scans with a variety of CT acquisition and reconstruction parameters; extensive validation of segmentation performance on 3 external datasets; "in silico" prospective clinical trial to compare the speed and reproducibility of their segmentation method to those of human experts.

In the revision, the authors have addressed most of the comments. Although some concerns around the moderate novelty were raised previously, this reviewer does not hold the view that technical innovation by itself is absolutely essential for a study to have an impact and move the field forward. This appears to be one of the first studies to report a fully automated and externally validated method for lung tumor segmentation. Upon further validation in prospective studies, this work could have implications for lung cancer screening/detection, radiotherapy planning, response evaluation, as well as for radiomics research.

I have a few additional comments which may help improve or clarify the manuscript.

1. The authors have evaluated quantitative performance in terms of image slice thickness, tumor size, physician-reported tumor complexity (depending on if PET is needed for segmentation – which is a bit subjective). Another important parameter that could impact segmentation performance is tumor location. See Wu, J., et al. *Nat Mach Intell* 3, 787–798 (2021). Can you evaluate the segmentation performance in different tumor subgroups such as peripheral tumors, or those with mediastinal or chest wall involvement?
2. The prognostic analysis is useful as it provides objective clinical evidence of automated segmentation (unlike manual delineation which is subjective and lacks a gold standard). Please also report C-index, and hazard ratio and p value from a univariable Cox regression – this does not depend on a cutoff point as in KM analysis. It may be useful to also look at the scatter plot for tumor volumes from manual and automated segmentations.
3. It is unclear whether all the scans are from treatment naïve patients. Since treatment can induce complex changes in the tumor and lung parenchyma (e.g., fibrosis), those cases may bring additional challenges for automated segmentation. In future work it might be useful to investigate your segmentation method specifically for response evaluation or post-treatment surveillance. Dedicated algorithms may need to be developed for this purpose. For example, PMID: 33767170. Could be worth discussing this point.

Point by point reply to reviewer's questions

Reviewer #1 (Remarks to the Author):

The authors have addressed a few concerns but did not address the major concerns including the method novelty and fair comparisons to other methods.

Response: We respectfully disagree with this statement and feel that our data provides the best data set currently addressing this topic in the literature. Our approach is also patented and extensively validated.

Authors' response that "3D architectures require quite some time to operate and computational power to run, whereas the 2D architectures are faster" is not true. 3D methods are just as fast as 2D methods. Several methods published in computer science conferences and journals (papers by Summers, Roth, Rueckert, Pheng-Ahn Heng, Kevin Zhou, etc) have shown feasibility of using 3D methods.

Response: The statement : "*3D architectures require quite some time to operate and computational power to run, whereas the 2D architectures are faster*" is not true " is, in our experience, not correct. For clarity, we do not claim that 3D methods are not feasible. In our response we have written that 3D architectures, require more time and computational power to handle. Other authors also endorse this point of view e.g. Zettler et al. 2021 mention: "*In this work, we focus on comparing 2D U-Nets vs. 3D U-Net counterparts. Our initial results indicate Dice improvements of about 6% at maximum. In this study to our surprise, liver and kidneys for instance were tackled significantly better using the faster and GPU-memory saving 2D U-Nets. For other abdominal key organs, there were no significant differences, but we observe highly significant advantages for the 2D U-Net in terms of GPU computational efforts for all organs under study.*" (<http://wscg.zcu.cz/WSCG2021/FULL/H61.pdf>)

Similarly, the novelty of image harmonization is claimed but is not really explained in detail as to why this is novel. The method description is standard intensity clipping based normalization.

Response: Our harmonization routine is more advanced than a "standard intensity clipping based normalization". It includes intensity normalization, spatial normalization, noise reduction, storage and calculations optimization, whereas "standard intensity clipping based normalization" is harmonizing only the intensity values. The image harmonization method has a step by step description in the methods section of the paper, lines: #419-436. The code is available for the reviewers, where the pre-processing part can be found in the Generator_v1.py file of "Automatic segmentation script" folder.

In summary, while there are some merits such as evaluation with respect to inter-raters, and use of multiple open-source datasets, method is very limited in novelty and the lack of fair comparisons to current state of the art methods, makes it difficult to assess the method's performance using fair settings.

Response: We respectfully disagree with this statement. The novelty of our paper has been recognized by the three other reviewers. We have extensively addressed this issue in the discussion section of the paper, lines: 314-330. Additionally, we have submitted a patent that covers our image data processing method (International Patent Application No. PCT/NL2020/050794 - "Pre-processing of CT images"), which has a favorable search report. Our work has also received an international award (ESTRO Jack Fowler award august 2021 (link1, link2)). Furthermore, we have performed a prospective *in-silico* clinical trial evaluating multiple automatic contouring related endpoints. The validation of automatic segmentation based purely on the manual segmentations is considered as the gold standard but still controversial because the Dr segmentations are not reproducible and have a large inter and intra Dr variability (fig 5b, fig 6). We described in Fig 4 and S9 an original approach of validation looking at the prognostic power of automated vs manual segmentations. In other words, we use a clinical outcome as endpoint to compare and evaluate the different segmentation methods. In summary, we believe our approach is novel for the image preprocessing and the validation.

Reviewer #2 (Remarks to the Author):

I would like to commend the authors for their detailed responses to each of the reviewer comments. I believe the changes have improved the readability of both the text and figures. Congratulations on this very interesting, well designed, and easy to follow study!

Response: We appreciate the reviewer's complementary evaluation.

Reviewer #3 (Remarks to the Author):

The authors addressed the questions and remarks well. I believe that the revised manuscript is of sufficient quality to be published in the journal.

Response: We thank the reviewer for the positive comments.

Reviewer #4 (Remarks to the Author):

This study presents a fully automated deep learning pipeline for the detection and volumetric segmentation of non-small cell lung cancer (NSCLC). Major strengths of the study include: use of a large multi-institution dataset of 1328 thoracic CT scans with a variety of CT

acquisition and reconstruction parameters; extensive validation of segmentation performance on 3 external datasets; “in silico” prospective clinical trial to compare the speed and reproducibility of their segmentation method to those of human experts.

In the revision, the authors have addressed most of the comments. Although some concerns around the moderate novelty were raised previously, this reviewer does not hold the view that technical innovation by itself is absolutely essential for a study to have an impact and move the field forward. This appears to be one of the first studies to report a fully automated and externally validated method for lung tumor segmentation. Upon further validation in prospective studies, this work could have implications for lung cancer screening/detection, radiotherapy planning, response evaluation, as well as for radiomics research.

Response: We thank the reviewer for the comments and valuable suggestions. We are very happy to hear that we share the same view on the novelty of our work and impact on the field.

I have a few additional comments which may help improve or clarify the manuscript.

1. The authors have evaluated quantitative performance in terms of image slice thickness, tumor size, physician-reported tumor complexity (depending on if PET is needed for segmentation – which is a bit subjective). Another important parameter that could impact segmentation performance is tumor location. See Wu, J., et al. Nat Mach Intell 3, 787–798 (2021). Can you evaluate the segmentation performance in different tumor subgroups such as peripheral tumors, or those with mediastinal or chest wall involvement?

Response: We thank the reviewer for the valuable input. We have updated the manuscript with the appropriate analysis and figures. We have added the description on the tumor location on the lines #552-556 of the updated manuscript and figure #7 in the updated supplementary materials. In short, on the test set we did not see a significant difference across the groups, however, on the external validation set DSC in the chest-wall involvement group was significantly ($2.161e-05$) higher than in the mediastinum group. Regardless of the tumor location median DSC across the groups was higher than 0.8 for both test and the external validation.

2. The prognostic analysis is useful as it provides objective clinical evidence of automated segmentation (unlike manual delineation which is subjective and lacks a gold standard). Please also report C-index, and hazard ratio and p value from a univariate Cox regression – this does not depend on a cutoff point as in KM analysis. It may be useful to also look at the scatter plot for tumor volumes from manual and automated segmentations.

Response: We thank the reviewer for such a thoughtful suggestion. We have incorporated the appropriate cox analysis in the manuscript. Table with the C-indexes, hazard ratios and p values can be found in the supplementary materials (supplementary table #3). We have also generated the scatter plots for automated vs manual tumor segmentations for both datasets. These figures can be found in the supplementary materials figure #10.

3. It is unclear whether all the scans are from treatment naïve patients. Since treatment can induce complex changes in the tumor and lung parenchyma (e.g., fibrosis), those cases may bring additional challenges for automated segmentation. In future work it might be useful to investigate your segmentation method specifically for response evaluation or post-treatment surveillance. Dedicated algorithms may need to be developed for this purpose. For example, PMID: 33767170. Could be worth discussing this point.

Response: We thank the reviewer for the comment and suggestion. We have provided this information in the line #404 and added mentioned prospects in the discussion lines#383-387. All images used to train or validate the developed model were planning/pretreatment scans. The treatment of cases depended on the clinical stage, and the local protocol, including surgery, chemotherapy, conventional chemo-radiotherapy and stereotactic radiotherapy.

REVIEWERS' COMMENTS

Reviewer #4 (Remarks to the Author):

All my comments have been thoroughly addressed by the authors.

I was also asked to comment on the authors' response to comments from reviewer 1. I can appreciate some of the novel aspects on image processing, which are clearly outlined by the authors in the manuscript. While these are not technical 'breakthrough' as expected from reviewer 1, what's important about this work is that authors have developed a practical working prototype for a fully automated tumor segmentation and performed such a comprehensive evaluation including an in silico prospective clinical trial. These strengths far outweigh the modest innovation perceived by reviewer 1. I'd like to commend the authors on this important and significant work.

REVIEWERS' COMMENTS

Reviewer #4 (Remarks to the Author):

All my comments have been thoroughly addressed by the authors. I was also asked to comment on the authors' response to comments from reviewer 1. I can appreciate some of the novel aspects on image processing, which are clearly outlined by the authors in the manuscript. While these are not technical 'breakthrough' as expected from reviewer 1, what's important about this work is that authors have developed a practical working prototype for a fully automated tumor segmentation and performed such a comprehensive evaluation including an in silico prospective clinical trial. These strengths far outweigh the modest innovation perceived by reviewer 1. I'd like to commend the authors on this important and significant work.

Response: We thank the reviewer for the positive comments.